# AgRP neuron activity enhances reward-related consummatory behaviors during energy deficit in mice

Daniela A. Cassano[1], Franco Barrile[1], Mirta Reynaldo[1], Gimena Fernandez[1], María P. Cornejo[1], María J. Tolosa[1], María F. Heredia[1], Nathalia Ferreira[2], Higor J. Fideles[2], Pablo N. De Francesco [1], Helgi B. Schiöth[3], Rodrigo Rorato [2] & Mario Perelló [1,3] ✉

Hunger enhances the consumption of rewarding foods, but the neurobiological basis of this adaptation remains unclear. We hypothesize that agouti-related protein (AgRP) neurons in the hypothalamic arcuate nucleus (ARH) promote the consumption of rewarding stimuli under calorie restriction, independent of caloric content. To test this, we study mice fed 40% of their average daily intake and exposed daily to the non-caloric sweetener saccharin before feeding. We show that calorie-restricted (CR) mice increase saccharin intake before each restricted feeding event and that this response requires ARH integrity. CR mice exhibit activation of AgRP neurons and their brain targets without significant changes in AgRP fiber density. Furthermore, satiated mice increase saccharin intake following chemogenetic activation of AgRP neurons, whereas CR mice with selective chemogenetic inhibition of AgRP neurons show reduced saccharin intake. Thus, we conclude that AgRP neuron activation enhances the consumption of a purely rewarding stimulus in CR mice. These findings contribute to our understanding of how the brain shapes food choices under conditions of energy deficit and could be important for managing food consumption during dieting or in eating disorders.

It is well recognized that hunger and low food availability enhance the appeal for some foods and their consumption. The adjustment of the rewarding value assigned to food stimuli and the enhancement of consummatory behaviors have been critical in evolution, ensuring survival by driving organisms to seek and consume specific types of foods—particularly those that are highly rewarding and energy-dense—during periods of energy deficit[1]. Such fundamental mechanisms controlling food intake are valuable in certain contexts, but may now be contributing to the excessive consumption of certain foods in modern obesogenic environments, thus exacerbating the current obesity epidemic. Despite its physiological relevance and potential implications, it is surprising that the neurobiological bases enhancing the consummatory responses to rewarding stimuli during energy deficit remain poorly understood.

Studies in mice have proven invaluable for uncovering the neural mechanisms underlying food intake regulation, notably highlighting the critical role of agouti-related protein (AgRP) neurons in the arcuate nucleus

of the hypothalamus (ARH) as key drivers of feeding behavior. Pharmacogenetic or optogenetic activation of AgRP neurons rapidly and potently induce voracious feeding in satiated mice[2,3]. Conversely, some, but not all[4], evidence shows that ablation of AgRP neurons in adult mice reduces food intake, even resulting in aphagia in some experimental paradigms[5,6]. AgRP neurons mediate the orexigenic effects of plasma peptide hormones, such as ghrelin[7,8], and increase not only food intake but also the willingness to work for and seek regular chow pellets in mice[3,9,10]. AgRP neurons are known to play a key role in the regulation of homeostatic feeding, they are activated under energy deficit conditions[11–13], and are required to trigger rebound hyperphagia when food becomes available after fasting[4,5]. In addition, AgRP neurons have been shown to modulate reward-related feeding behaviors. For instance, a recent study found that chemogenetic activation of AgRP neurons increases the consumption of a lipid emulsion and sucrose in satiated mice[14]. In line with these behavioral observations, chemogenetic activation of AgRP neurons in *ad libitum* fed mice was shown to enhance

[1]Laboratory of Neurophysiology of the Multidisciplinary Institute of Cell Biology [IMBICE, Argentine Research Council (CONICET) and Scientific Research Commission, Province of Buenos Aires (CIC-PBA), National University of La Plata], La Plata, Argentina. [2]Department of Biophysics, Paulista Medical School, Federal University of São Paulo (UNIFESP), São Paulo, Brazil. [3]Department of Surgical Sciences, Functional Pharmacology and Neuroscience, University of Uppsala, Uppsala, Sweden. ✉e-mail: mperello@imbice.gov.ar

dopamine release in the nucleus accumbens in response to food consumption[15]. Based on this background, we hypothesized that the activation of AgRP neurons mediates the well-established enhancement of consummatory responses to rewarding stimuli during energy deficit. To test this hypothesis, we employed an experimental paradigm in which mice were calorie-restricted (CR; i.e., fed 40% of their average daily food intake) and exposed daily to a non-caloric saccharin solution prior to feeding. We assessed saccharin consumption in mice with selective manipulations of the ARH nucleus or AgRP neurons and found evidence indicating that the activation of AgRP neurons is required to enhance saccharin intake during calorie restriction and is sufficient to induce saccharin intake in satiated conditions.

## Material and methods

### Animals
This study used 2- to 5-month-old male mice from the IMBICE or the Biophysics Department of UNIFESP. Mice were housed at 21 ± 1 °C with a 12-h light/dark cycle (7:00 to 19:00) and had *ad libitum* access to chow and water, unless specified. We have complied with all relevant ethical regulations for animal use. All protocols were prepared prior to the study and were approved and registered by the Institutional Animal Care and Use Committee at each institution. Experimental groups included: (1) wild-type (WT) C57BL/6 J mice, (2) ARH-ablated and ARH-intact mice, generated by subcutaneous injections of monosodium glutamate (2.5 mg/g body weight (BW)) or 10% saline, respectively, to 4-day-old pups[16]; (3) neuropeptide Y (NPY)-hrGFP mice, which express humanized *Renilla* GFP under the control of the NPY promoter (Jackson Laboratory, #006417)[17]; (4) AgRP-Cre mice, which express Cre under the control of the AgRP promoter (Jackson Laboratory, #012899)[18]; (5) Ai14 mice (Allen Institute, #007908), which express tdTomato after Cre recombination[19]; (6) Gi mice (R26-LSL-Gi-DREADD mice, Jackson Laboratory, #026219), which express HA-hM4Di-pta-mCitrine after Cre recombination[20]; (7) AgRP-Gi mice, generated by crossing AgRP-Cre with Gi mice; 8) AgRP-tdTomato mice were generated by crossing AgRP-Cre with Ai14 mice. All genetically modified mice were backcrossed for more than 10 generations onto a C57BL/6 J genetic background.

### Experimental design
The experimental design is shown in Fig. 1a. Mice were individually housed in cages enriched with nesting material and shelters four days prior to the experiment, with *ad libitum* access to food and water to estimate their daily intake. The night before the experiment, they were given access to two drinking bottles: one containing a 0.1% sodium saccharin (Parafarm) solution and the other containing water. On the first experimental day, food was removed at 10:00, and mice were given 4-hour access to the saccharin solution while retaining access to water, the positions of the bottles (left or right side of the cage) were randomly assigned for each animal. Afterward, they were weighed and randomly assigned to experimental groups: either fed *ad libitum* (fed mice) or given 40% of their average daily intake (CR mice). This daily routine—saccharin exposure from 10:00 to 14:00 followed by either *ad libitum* or 40% feeding, depending on the group—was repeated for four consecutive days. Saccharin and water intake were calculated separately as raw values, and saccharin preference was calculated as (saccharin intake/total intake) × 100%, where total intake refers to the combined volume of saccharin and water consumed. On the fifth experimental day, both fed mice (which had been eating *ad libitum*) and CR mice (which had been receiving 40% of their average daily intake for the previous four days) were perfused at 10:00, prior to saccharin exposure.

### DREADDs experiments
To pharmacogenetically activate AgRP neurons, anesthetized WT or AgRP-Cre mice were bilaterally injected in the ARH (posterior = 1.4 mm, lateral = ±0.3 mm, and ventral = 5.85 mm from Bregma) with 300 nL of pAAV8-hSyn-DIO-hM3D(Gq)-mCherry (Addgene; viral titer ~1.8 × 10^13 vg/mL), as previously detailed[21]. Mice were allowed to recover

for 2 weeks after surgery. For behavioral experiments, individually housed WT and AgRP-Cre mice injected with the hM3Dq-mCherry AAV virus were daily habituated to handling by administering saline solution for 3 consecutive days. To assess DREADD activation-induced food intake, all mice were intraperitoneally (IP)-injected with clozapine N-oxide (CNO, ENZO, 1 μg/g BW) at 10:00, and food intake was measured at 4 h after treatment as the difference in pellet weight from time zero. To assess saccharin consumption, AgRP-Cre mice injected with the hM3Dq-mCherry AAV virus were provided overnight access to two drinking bottles: one containing a 0.1% sodium saccharin solution and the other containing water. Four days later, *ad libitum* fed mice were IP-injected with vehicle or CNO randomly and, 30 min later, exposed to a saccharin solution alone for 4 h. After another 4 days, mice underwent a crossover protocol with IP injections of vehicle or CNO, followed by a 4-h exposure to the saccharin solution alone, as described above. Finally, all mice were injected with CNO and perfused 2 h later, as described below, without access to saccharin or food. The accuracy of targeting the ARH in AgRP-Cre mice with the hM3Dq-mCherry AAV virus was estimated based on the induction of food intake in response to CNO along with the post-mortem assessment of mCherry expression and CNO-induced c-Fos levels in the ARH.

To pharmacogenetically inhibit AgRP neurons, we used AgRP-Gi mice. To test whether CNO treatment functions as predicted in this mouse model, individually housed WT, AgRP-Cre, Gi, and AgRP-Gi mice were first IP-injected with either vehicle or CNO as described above and, 1 h later, with ghrelin (60 pmol/g BW; Global Peptides, cat. PI-G-03). Immediately after ghrelin administration, mice were exposed to a pre-weighed chow pellet, and food intake was manually recorded after 2 h. Two weeks later, some of them were randomly re-treated with vehicle or CNO, followed by ghrelin as described above, and perfused 2 h later without food. In another experiment, WT, AgRP-Cre, Gi, and AgRP-Gi mice underwent the calorie restriction and daily saccharin access protocol (see "Experimental Design"), receiving daily IP injections of vehicle or CNO (1 μg/g BW), randomly assigned and administered 30 min before each saccharin access period. After 4 consecutive days, mice were provided with 40% of their average food intake and perfused the following day, 2 h after CNO treatment, without access to saccharin or food.

### Quantitative PCR
As previously detailed[22], AgRP mRNA levels were quantified using RT-qPCR in ARH microdissection from WT *ad libitum* fed or CR mice. Fold change from *ad libitum* fed values was determined using the Pfaffl method[23]. Primers sequences for AgRP were sense: 5′-CTGAGTTGTGTTCTGCTGTT-3′, antisense: 5′-GACTTAGACCTGGGAACTCT-3′ (GenBank Accession No. NM_007427.3). Primers sequences for ribosomal protein L19 (housekeeping) were sense: 5′-AGCCTGTGACTGTCCATTCC-3′, antisense: 5′-TGGCAGTACCCTTCCTCTTC-3′ (GenBank Accession No. NM_009078.2).

### Immunohistochemistry (IHC)
Anesthetized mice were perfused with formalin, and their brains were extracted and post-fixed as previously detailed[24]. Frozen brains were coronally cut at 40 μm into four equal series on a sliding cryostat (Leica). For chromogenic IHC, sections were washed in phosphate buffer saline (PBS), treated with 0.5% $H_2O_2$, blocked with 20% horse serum in 0.25% Triton-X in PBS, and incubated for 48 h at 4 °C with rabbit anti-c-Fos (Abcam, ab214672, 1:40000, RRID:AB_2939046). After washing, sections were incubated for 1 h with biotinylated anti-rabbit antibody (VectorLabs, BA1000, 1:3000, RRID:AB_2313606) and processed using an avidin-biotin complex (Vectastain Elite ABC kit, VectorLabs cat. 6100, RRID:AB_2336819) and diaminobenzidine for brown staining. Sections were mounted and coverslipped with mounting medium. For fluorescent IHC, unstained sections or sections subjected to IHC against c-Fos were blocked with 20% horse serum in 0.25% Triton-X PBS, and incubated for 48 h at 4 °C with guinea pig primary antibody anti-c-Fos (Synaptic Systems, 226308, 1:25000, RRID:AB_2905595), rabbit antibody anti-AgRP (Phoenix,

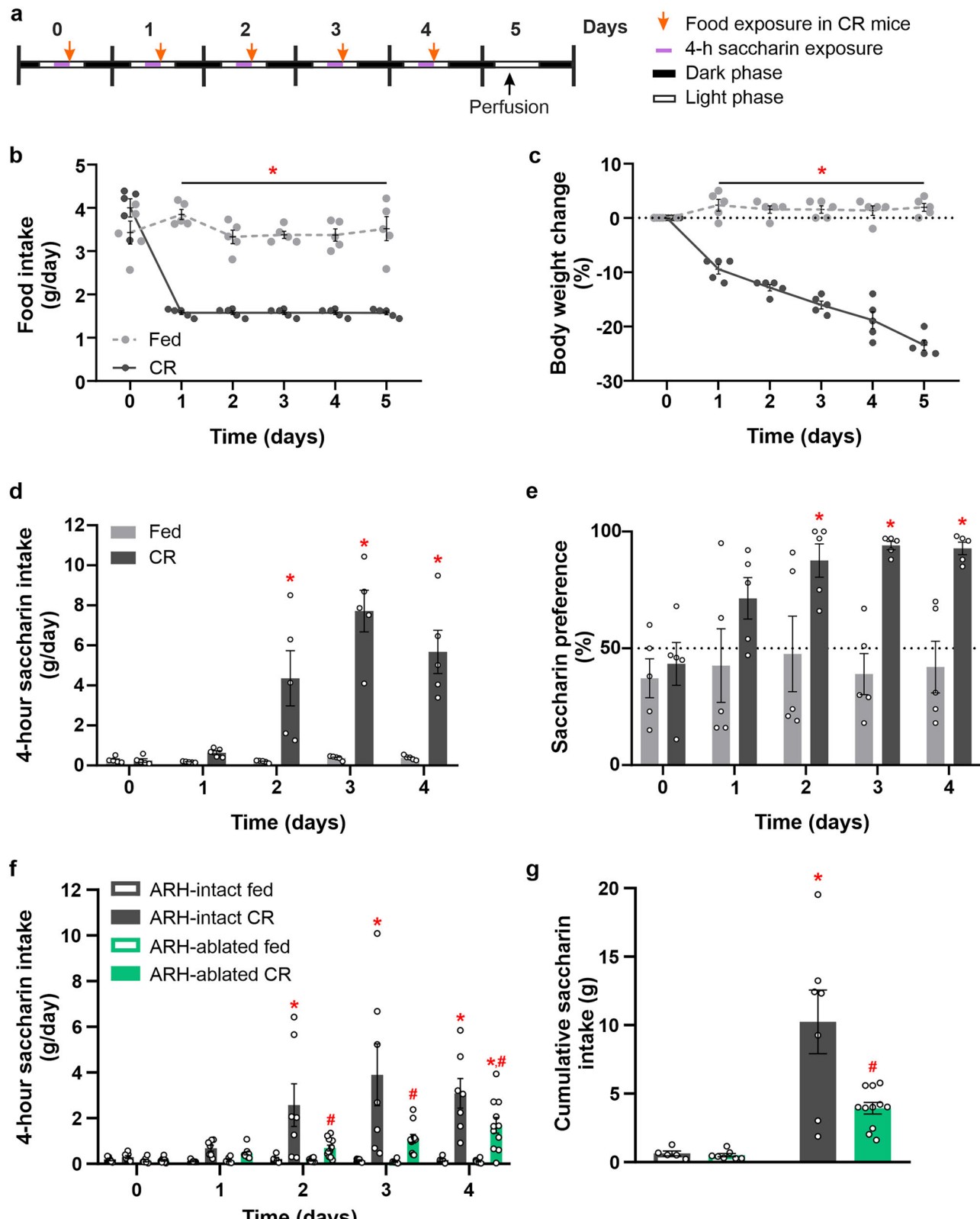

H-003-57, 1:2000, RRID: AB_2313909), or anti-hrGFP (Stratagene, 240142, 1:1000, RRID: AB_2314658), or chicken anti-eGFP (Origene, TA100022, 1:5000) to detect mCitrine. After washing, sections were incubated for 2 h with anti-rabbit Alexa Fluor 594- or 488-conjugated (Thermo Fisher, A-21207, RRID: AB_141637 or A-32790, RRID:AB_2762833; 1:1000), anti-guinea pig Alexa 488-conjugated (Thermo Fisher, A-11073; RRID:AB_

2534117, 1:1000) or anti-chicken Alexa 488-conjugated (Invitrogen, A11039, RRID:AB_2534096, 1:1000) secondary antibodies. Finally, sections were mounted with bisBenzimide-containing medium.

To validate AgRP-cre line, AgRP-tdTomato mice ($n = 3$) were intra-cerebroventricularly (ICV) injected with colchicine (8 μg/mouse, Sigma, C9754) in the lateral ventricle (coordinates: posterior = 0.3 mm, lateral =

**Fig. 1 | The integrity of the ARH is required for the enhancement of saccharin intake induced by calorie restriction. a** The experimental timeline summarizes the design used in the current study. Male mice were calorie restricted to 40% of their basal intake for 5 days and had restricted access to a saccharin solution (0.1%) for 4 h every day while water was *ad libitum* available. During the limited access period, chow was removed from home cages. Mice were perfused on day 5, prior to any access to saccharin or food. **b–e** Show food intake (**b**, *p*-time x group < 0.001, Cohen's *f* = 1.984), body weight change, (**c**, *p*-time x group < 0.001, Cohen's *f* = 3.084), saccharin intake (**d**, *p*-time x group < 0.001, Cohen's *f* = 1.49) and saccharin preference (**e**, *p*-time x group < 0.001, Cohen's *f* = 1.013), of wild-type (WT) mice that were maintained with *ad libitum* access to regular chow or calorie restricted (CR) for 5 days (*n* = 5 mice per group). **f, g** Show saccharin intake (**f**, *p*-time x treatment = 0.060, Cohen's *f* = 0.341; *p*-time x group < 0.001, Cohen's *f* = 0.724; *p*-treatment x group = 0.0146, Cohen's *f* = 0.464; *p*-time x treatment x group = 0.070, Cohen's *f* = 0.334) and cumulative saccharin intake (**g**, *p*-group x treatment = 0.015, Cohen's *f* = 0.515) of WT ARH-intact or ARH-ablated mice that were maintained with *ad libitum* access to regular chow (*n* = 5 and 7 mice per group) or CR (*n* = 7 and 11 mice per group). Data represent the mean ± SEM, with error bars representing the SEM, and were compared by two-way ANOVA (**b–e**, **g**) or three-way ANOVA (**f**). *, *p* < 0.05 vs. same treatment different group; #, *p* < 0.05 different treatment same group.

1.0 mm, ventral = 2.3 mm, from Bregma), as described previously[24]. As it is known that colchicine treatment can result in severe discomfort, this treatment was kept as short as possible and extra care was taken to avoid suffering. Mice were individually housed and monitored continuously to assess general behavior (activity, grooming, posture), food and water intake, and signs of distress (piloerection, vocalization, abnormal gait, hunched posture). No animals met the humane endpoint criteria—such as sustained loss of righting reflex, >20% BW loss, severe dehydration, lack of grooming, or persistent abnormal behavior—and all were perfused 48 h after colchicine injection. Brain sections were subjected to a fluorescent IHC anti-AgRP as described above.

### Quantitative imaging analysis

Brain regions were identified and delineated using a mouse brain atlas[25]. Bright-field images of chromogenic c-Fos labeling were acquired using a Nikon Eclipse 50i microscope equipped with a DS-Ri1 Nikon camera. Fluorescent images were obtained with a Zeiss AxioObserver D1 microscope featuring an Apotome.2 structured illumination module and an AxioCam 506 camera. Emission filters used included 400–550 nm for Alexa Fluor 488, 570–640 nm for Alexa Fluor 594, tdTomato and mCherry, and 420–470 nm for Hoechst. Cells positive for c-Fos (c-Fos+ cells) were identify by the presence of brown or fluorescent nuclear precipitate. Cells positive for AgRP, hrGFP or tdTomato were identified by the presence of fluorescent signal in the cytoplasma and referred to as AgRP + , hrGFP+ or tdTomato+ cells, respectively. Single- and double-labeled cells in the images were manually identified and quantified in a blinded manner by a trained operator using Fiji software. Additionally, the fluorescent intensity and fluorescent area in brain regions labeled for AgRP+ signal was assessed using Fiji, as previously detailed[26], and the script is available in the Zenodo repository (https://doi.org/10.5281/zenodo.3541615).

### Statistics and reproducibility

Statistical analyses were performed using GraphPad Prism 8.0. The number of mice used in all experiments was informed by pilot studies and exceeded the minimum required to detect medium-to-large effect sizes (Cohen's *f* > 0.25 for ANOVA or Cohen's *d* > 0.5 for t-tests) with adequate power for hypothesis testing. Data normality was assessed with the D'Agostino & Pearson test. Normally distributed data were expressed as mean ± SEM and analyzed using one-tailed unpaired or one-sample Student's t-tests or two- or three-way ANOVA. Significance was set at *p* < 0.05. All experiments were performed in at least two independent biological replicates, with consistent results across replicates. Where applicable, data were collected and analyzed in a blinded manner. As described above, potential confounders were minimized through randomization. No mice were excluded from the analysis, except in cases where saccharine intake values exceeded ±2 standard deviations from the group mean, and the data otherwise met the assumptions of normality. Specifically, 3 mice were excluded from the vehicle-treated AgRP-Gi group and 3 from the CNO-treated AgRP-Gi group. Detailed descriptions of sample sizes, statistical tests, and power analysis are provided in the figure legends and throughout the text.

## Results

### Calorie restriction enhances saccharin intake in an ARH-dependent manner

To estimate the consummatory response to a non-caloric, palatable stimulus, we used a saccharin solution, a sweetener that is highly rewarding to mice[27]. This was evidenced by their consumption during the overnight priming exposure: 0.76 ± 0.39 g of water vs 4.41 ± 0.44 g of saccharin (t-test, *p* < 0.001, Cohen's *d* = 2.72), indicating an ~85% preference for saccharin over water. To first assess if consummatory responses to the saccharin solution were enhanced during energy deficit, WT mice were daily exposed to 40% of their *ad libitum* food intake during the light phase for 5 days, resulting in a significant decrease in BW compared to *ad libitum*-fed mice, and were also daily exposed to a saccharin solution for 4 h before their scheduled feeding (Fig. 1a–c). In *ad libitum*-fed mice, we observed that both daily water intake (measured as 0.65 ± 0.23, 0.45 ± 0.17, 0.48 ± 0.18, 0.70 ± 0.17, and 0.80 ± 0.26 g for experimental days 0 to 4, one-way ANOVA, *p* = 0.504) and daily saccharin intake (Fig. 1d) remained low and unchanged during the 4 hours of saccharin exposure, with saccharin preference averaging around 50% across all experimental days (Fig. 1e). Conversely, daily saccharin intake increased in CR mice from experimental day 2 to day 4 (Fig. 1d). Daily water intake in CR mice remained low and stable (0.48 ± 0.23, 0.27 ± 0.09, 0.28 ± 0.15, 0.42 ± 0.08, and 0.36 ± 0.10 g on experimental days 0 to 4, respectively, one-way ANOVA, *p* = 0.374), leading to a marked increase in daily saccharin preference over time (Fig. 1e).

Since the ARH plays a key role in regulating consummatory responses, we compared saccharin consumption between ARH-intact and ARH-ablated mice exposed to the protocol described above. Food intake and BW of ARH-intact and ARH-ablated mice were similar throughout the experiment (Supplementary Fig. 1). However, saccharin intake showed a significant increase in CR ARH-ablated mice only on day 4 of the protocol, compared to *ad libitum*-fed ARH-ablated mice, with their intake representing ~27%, ~28%, and ~55% of the saccharin intake observed in CR ARH-intact mice on days 2, 3, and 4, respectively (Fig. 1f). Consequently, total saccharin intake over the experiment was significantly smaller in CR ARH-ablated mice compared to CR ARH-intact mice (Fig. 1g). Thus, increased consummatory responses to a palatable stimulus in CR mice require an intact ARH.

### Calorie restriction results in the activation of AgRP neurons

To gain insights into the neurobiological basis underlying the increased consummatory responses to a palatable stimulus in CR mice, we analyzed c-Fos levels in the ARH of *ad libitum*-fed and CR mice perfused on the fifth day of the protocol, prior to saccharin consumption, and found that the number of c-Fos+ cells increased in the ARH of CR mice (Fig. 2a, b). Since ARH AgRP neurons are key players during energy deficit conditions[12,28], we performed IHC against AgRP to estimate AgRP peptide levels and found that the AgRP+ signal was higher in the ARH of CR mice (Fig. 2c, d). In *ad libitum*-fed mice, AgRP+ signal displayed a granular pattern, enriched around Hoechst+ cell nuclei, and was abundantly present throughout the ARH. In CR mice, AgRP+ signal more clearly labeled cell bodies located in the ventromedial region of the ARH. We also measured AgRP mRNA levels in the ARH and confirmed

**Fig. 2 | AgRP neurons are activated during calorie restriction. a** Bar graph displaying the quantitative analysis of the number of c-Fos+ cells in the hypothalamic arcuate nucleus (ARH) of wild-type (WT) mice *ad libitum* fed or calorie restricted (CR) (*n* = 5 mice per group, *p* = 0.005, Cohen's *d* = 2.267). **b** Representative photomicrographs of the coronal section of ARH subjected to chromogenic immunohistochemistry against c-Fos (brown). **c** Bar graphs displaying the quantitative analysis of the mean fluorescence intensity of AgRP+ in the ARH of WT mice *ad libitum* fed or CR (*n* = 4 and 6 mice per group, *p* = 0.023, Cohen's *d* = 1.10). **d** Representative photomicrographs of coronal section of ARH subjected to fluorescent immunohistochemistry against AgRP (red). **e** Pearson correlation between cumulative saccharin intake and AgRP cells positive for c-Fos (%) in CR mice (*n* = 6 mice). **f** Representative photomicrographs of the coronal section of the ARH of WT mice subjected to chromogenic immunohistochemistry against c-Fos (cyan) and fluorescent immunohistochemistry against AgRP (red). **g** Bar graphs displaying the quantitative analysis of the hrGFP+ cells positive for c-Fos (%) in the ARH of NPY-hrGFP mice that were maintained with *ad libitum* access to regular chow or CR for 5 days (*n* = 3 and 6 mice per group, *p* = 0.024, Cohen's *d* = 4.97). **h** Representative photomicrographs of the coronal section of ARH of NPY-hrGFP mice subjected to chromogenic immunohistochemistry against c-Fos (magenta) and fluorescent immunohistochemistry against hr-GFP (green). Colocalization is shown with white arrowheads and c-Fos+ signal with blue arrowheads. Scale bar: 50 µm low magnification and 25 µm high magnification. Data represent the mean ± SEM, with error bars representing the SEM, and were compared by t-test. *, *p* < 0.05 vs. *ad libitum* fed.

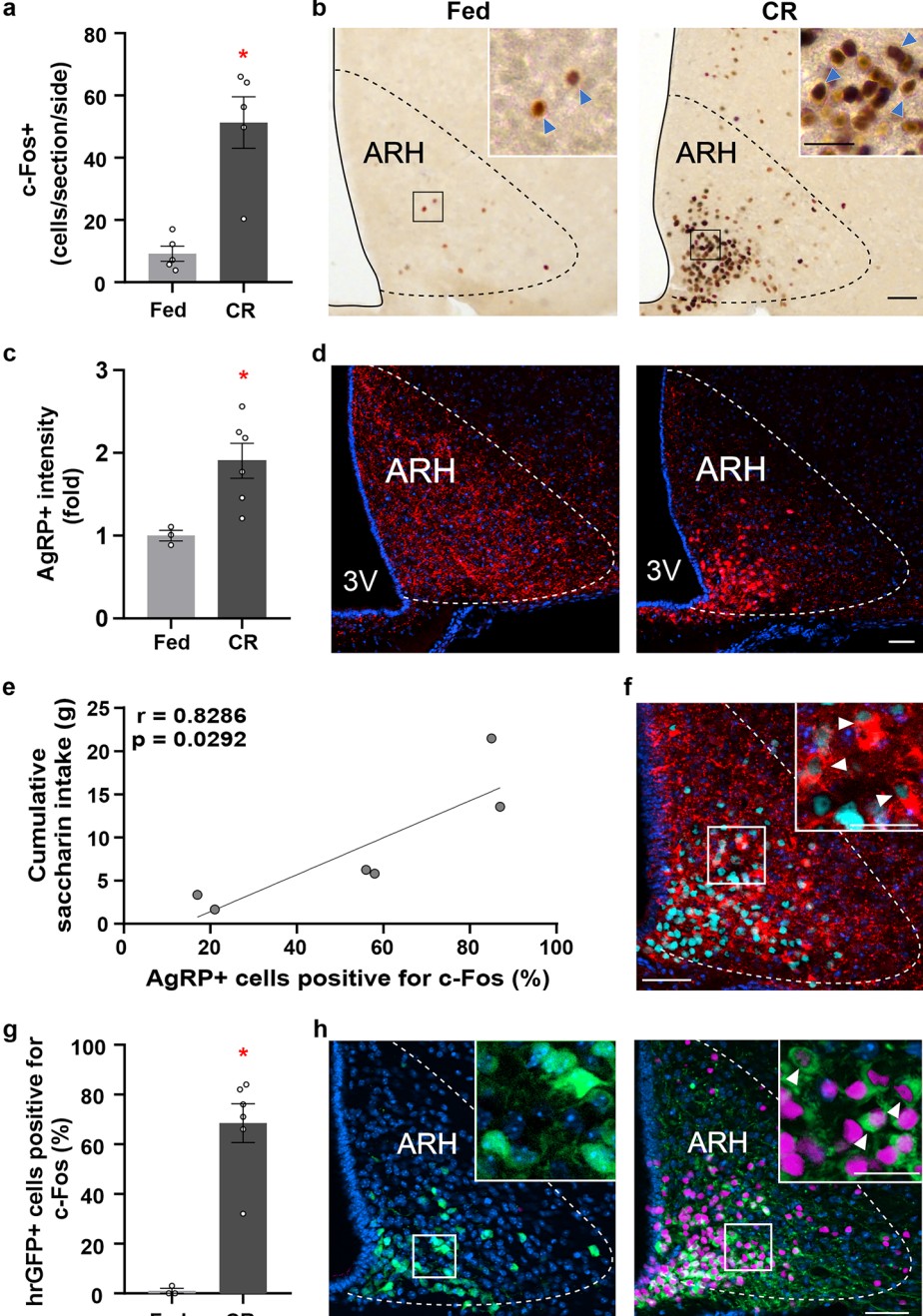

an increase in CR mice compared to *ad libitum*-fed mice (3.9 ± 1.1 vs. 1.3 ± 0.5, respectively, t-test, *p* = 0.043, Cohen's *d* = 1.39). To estimate AgRP neuron activation, we performed double IHC against c-Fos and AgRP. We found that AgRP+ cells positive for c-Fos were only evident in CR mice (54 ± 12% cells/section/side, one-sample t-test *p* = 0.007, Cohen's *d* = 1.79), and notably, the fraction of AgRP+ cells positive for c-Fos in each mouse correlated with total saccharin intake over the experiment (Fig. 2e, f). To more confidently visualize AgRP neurons in both experimental groups, we also studied c-Fos levels in the ARH of NPY-hrGFP mice, in which most hrGFP+ cells are AgRP neurons[17,29]. We found that the fraction of hrGFP+ cells positive for c-Fos was higher in the ARH of CR NPY-hrGFP mice compared to *ad libitum*-fed NPY-hrGFP mice (Fig. 2g, h). Also, most c-Fos+ cells in the ARH of CR NPY-hrGFP mice were hrGFP+ cells (8 ± 8 vs. 45 ± 2% of all c-Fos+ cells, t-test *p* < 0.001, Cohen's *d* = 4.15), Thus, most AgRP neurons are activated in CR mice, and most activated cells of the ARH are AgRP neurons.

We also performed a neuroanatomical analysis of c-Fos+ cells in brain targets of AgRP neurons that are known to mediate consummatory responses, including the bed nucleus of the stria terminalis (BNST), paraventricular hypothalamic nucleus (PVH), paraventricular thalamic nucleus (PVT), and lateral hypothalamic area (LHA)[30]. The analysis revealed that the number of c-Fos+ cells only increased in the PVH and BNST and showed a tendency to increase in the LHA and PVT of CR mice (Fig. 3 and Supplementary Fig. 2). We have shown that the projections of AgRP neurons toward other hypothalamic areas are remodeled during fasting, and such remodeling correlates with changes in food consumption[26]. Thus, we quantified the area with AgRP+ signal, which reflects the density of fluorescent fibers, in different brain areas innervated by AgRP neurons. We found that the fluorescent area of the AgRP+ signal decreased in the BNST, whereas it was unaffected in the PVH, PVT, and LHA, showing no difference between CR and *ad libitum*-fed WT mice. (Fig. 3 and Supplementary Fig. 2).

**Fig. 3 | Calorie restriction induces c-Fos expression in AgRP neuronal targets. a, c** Bar graph displaying the quantitative analysis of the number of c-Fos+ cells in the paraventricular hypothalamic nucleus (PVH; $p = 0.011$, Cohen's $d = 0.99$) and bed nucleus of the stria terminalis (BNST; $p < 0.001$, Cohen's $d = 3.22$), respectively, of wild-type (WT) mice *ad libitum* fed ($n = 5$ and 4 mice per group) or calorie restricted (CR) ($n = 5$ and 4 mice per group). **b, d** Representative photomicrographs of the coronal section of PVH and BNST, respectively, subjected to chromogenic immunohistochemistry against c-Fos (brown signal, blue head arrows). **e, g** Bar graph displaying the quantitative analysis of the fluorescent area in the PVH ($p = 0.801$) and BNST ($p = 0.0013$, Cohen's $d = 3.92$), respectively, of WT mice *ad libitum* fed or CR ($n = 4$ and 6 mice per group). **f, h** Representative photomicrographs of the coronal section of PVH and BNST, respectively, of WT mice subjected to fluorescent immunohistochemistry against AgRP (red). Scale Bar: 50 μm low magnification and 25 μm high magnification. Data represent the mean ± SEM, with error bars representing the SEM. Data were compared by t-test. *, $p < 0.05$ vs. *ad libitum* fed. aca, anterior commissure, anterior part.

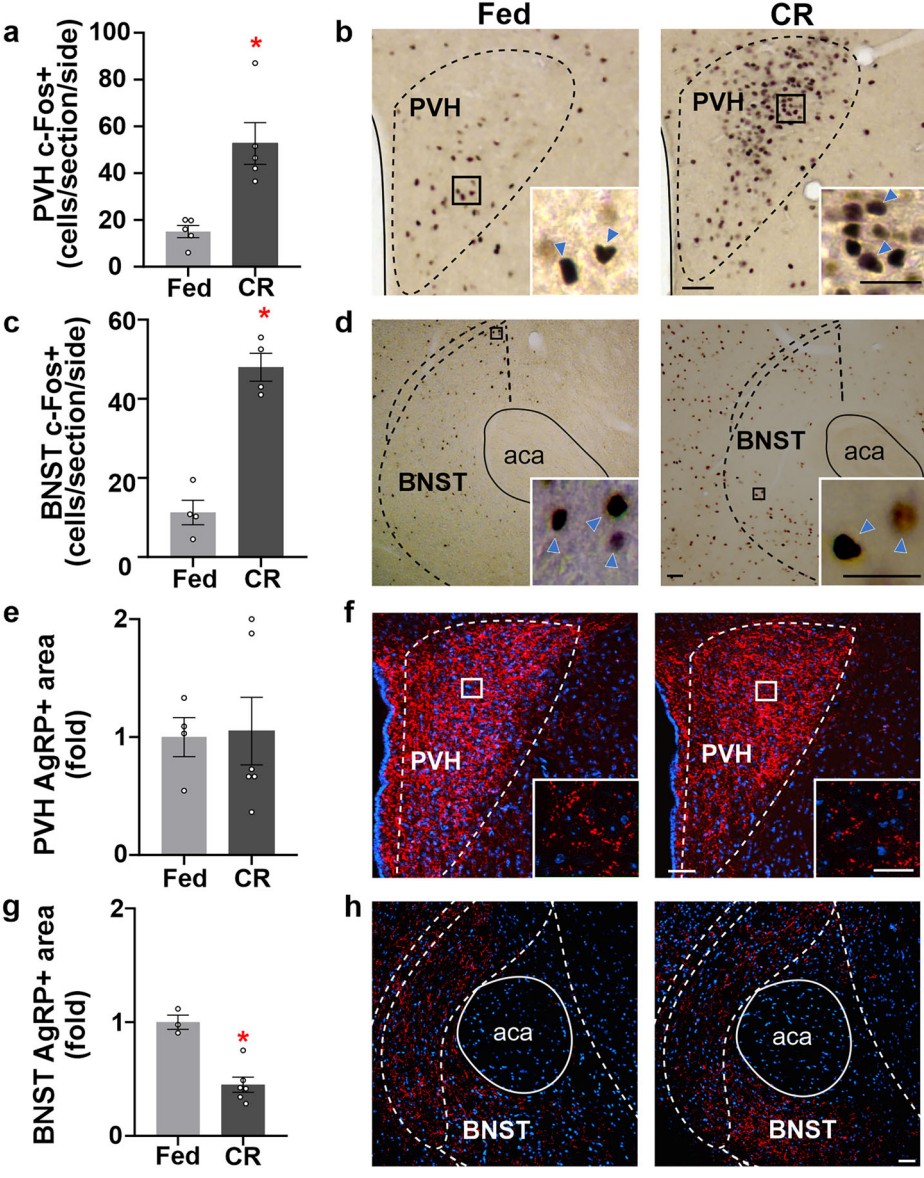

## Pharmacogenetic activation of AgRP neurons is sufficient to induce saccharin intake in *ad libitum* fed mice

Next, we tested whether AgRP neuron activation is sufficient to enhance saccharin intake in satiated mice by employing DREADD technology in AgRP-Cre mice. We first confirmed that Cre recombinase was active in 90.8 ± 0.3% of all AgRP neurons in these mutant mice and minimally active in other brain areas, as determined by analyzing the tdTomato+ cells in mice generated by crossing AgRP-Cre mice with a reporter line (Supplementary Fig. 3). Next, we performed stereotactic injections of hM3Dq-mCherry AAV virus into the ARH of WT or AgRP-Cre mice, resulting in the expression of the excitatory hM3Dq protein in AgRP neurons only of AgRP-Cre mice (Fig. 4a). Of note, CNO treatment increased c-Fos levels in the ARH of AgRP-Gq mice, but not in WT mice injected with hM3Dq-mCherry AAV virus into the ARH (Fig. 4a). Consistent with previous reports[3], we found that CNO treatment increases 4-hour food intake in AgRP-Gq mice with mCherry targeted to the ARH, but not in WT mice injected with the hM3Dq-mCherry AAV virus into the ARH, or in AgRP-Cre mice where hM3Dq-mCherry AAV virus injections failed to target the ARH (1.50 ± 0.07, 0.29 ± 0.10 and 0.51 ± 0.23 g, respectively, one-way ANOVA, $p < 0.001$, Cohen's $f = 5.87$). In crossover studies, we found that CNO treatment, compared to vehicle treatment, increased saccharin intake

in targeted AgRP-Gq mice with mCherry expression restricted to the ARH (Fig. 4b). Conversely, CNO treatment did not affect saccharin intake in AgRP-Cre mice in which hM3Dq-mCherry AAV virus injections failed to successfully target the ARH (2.59 ± 0.57 vs. 2.56 ± 0.60 g for CNO and vehicle conditions, respectively; paired t-test, $p = 0.966$).

## Calorie restriction-induced enhancement of saccharin intake requires AgRP neuron activation

To test whether AgRP neuron activation is required to enhance saccharin intake during energy deficit, we used DREADD technology to selectively inhibit AgRP neurons each day prior to saccharin exposure in CR mice. To generate mice in which AgRP neurons could be specifically inhibited, AgRP-Cre mice were crossed with Gi mice to produce AgRP-Gi mice, in which Cre-mediated removal of the floxed-STOP cassette induces selective expression of the inhibitory hM4Di-mCitrine in AgRP neurons. Here, we tested the specificity of hM4Di-mCitrine expression by analyzing mCitrine signal in the brain of CR mice and found it to be enriched in the ventromedial ARH, overlapping with AgRP signal in AgRP-Gi mice, whereas no such signal was observed in AgRP-Cre or Gi control mice (Supplementary Fig. 4). Also, we tested the effects of ghrelin in these mice pretreated with either vehicle or CNO to assess the functional efficacy of CNO-induced

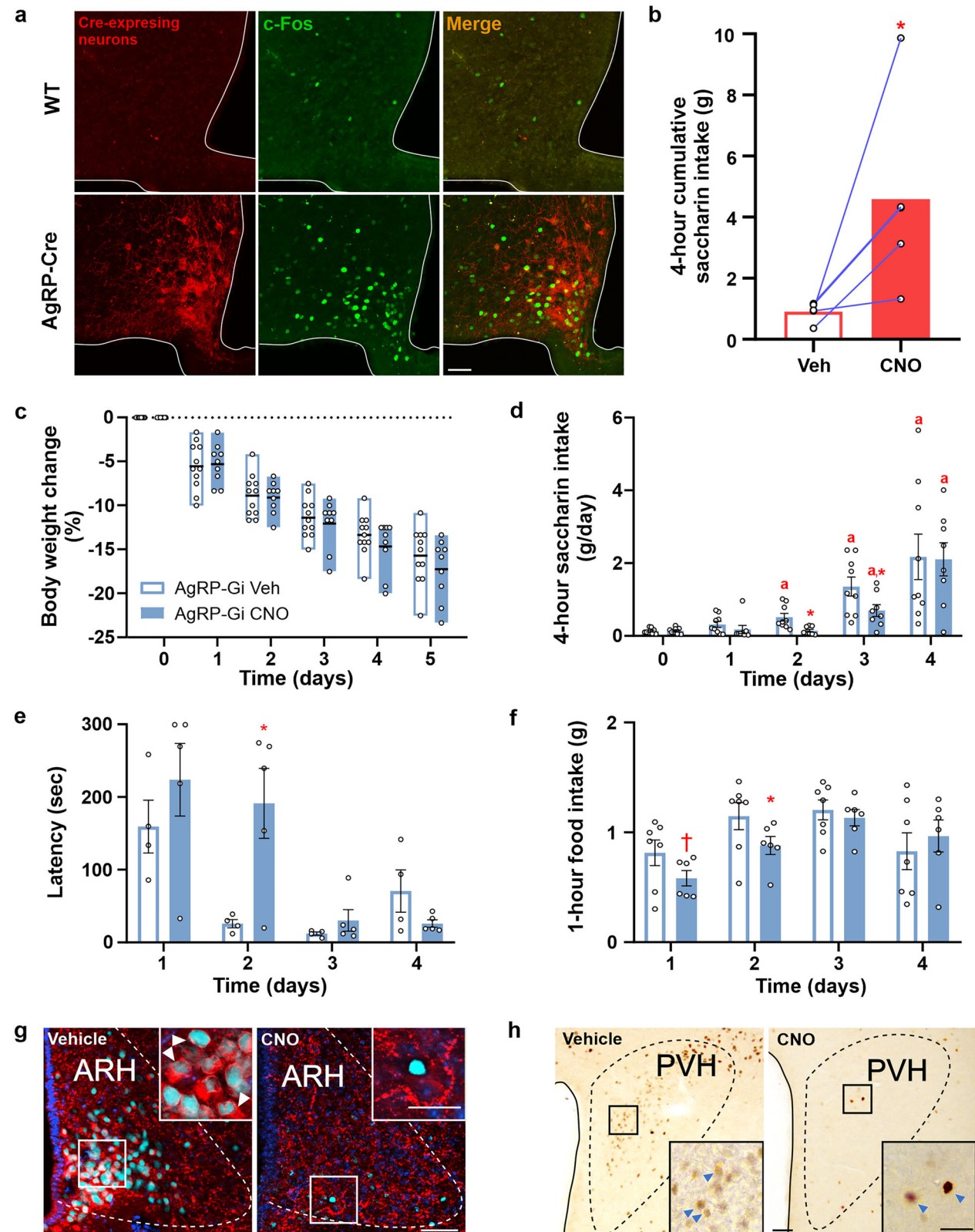

inhibition of AgRP neurons, given that systemic ghrelin treatment primarily increases food intake by acting on these neurons[7]. We found that ghrelin increased food intake and c-Fos labeling in the ARH of vehicle- or CNO-treated WT, AgRP-Cre, and Gi mice, which were grouped together and referred to as "control" mice (Supplementary Fig. 4). Conversely, ghrelin

increased food intake and c-Fos labeling in the ARH of vehicle-treated AgRP-Gi mice, but not in CNO-treated ones (Supplementary Fig. 4).

Next, we subjected control and AgRP-Gi mice to the aforementioned calorie restriction and daily saccharin access protocol (Fig. 1a). In control mice, we found that their behavior in response to the protocol resembled

**Fig. 4 | AgRP neurons are required for the enhanced saccharin intake during calorie restriction. a** Representative photomicrographs of coronal section of hypothalamic arcuate nucleus (ARH) of wild-type (WT) or AgRP-Cre mice stereotaxically injected with pAAV8-hSyn-DIO-hM3D(Gq)-mCherry (red). Mice were perfused after CNO treatment and brain sections were subjected to a fluorescent immunohistochemistry against c-Fos (green). **b** Bar graphs display saccharin intake for 4 h after vehicle or CNO treatment 30 min before access to saccharin solution in AgRP-Gq mice in a crossover protocol ($n = 5$ mice per group, paired t-test, $p = 0.029$, Cohen's $d = 1.21$). **c** Displays body weight change of CR AgRP-Gi mice treated with vehicle or CNO ($n = 11$ and 9 mice per group, two-way ANOVA, $p$-time x treatment = 0.413; $p$-treatment = 0.515; $p$-time < 0.001, Cohen's $f = 3.96$). **d** Shows saccharin intake of CR AgRP-Gi mice that were treated with vehicle or CNO before saccharin exposure ($n = 9$ and 8 mice per group; two-way ANOVA: $p$-time x treatment = 0.703; $p$-treatment = 0.263; $p$-time < 0.001; Cohen's $f = 1.183$. Dunnett's post hoc tests: days 1, 2, 3, and 4 vs. day 0 in the vehicle-treated group: $p = 0.203$, 030, 0.006, and 0.036; in the CNO-treated group: $p = 0.994$, 0.999, 0.042, and 0.010. Vehicle- vs. CNO-treated groups: t-test on day 1, $p = 0.165$; day 2, $p = 0.003$, Cohen's $d = 1.55$; day 3; $p = 0.028$, Cohen's $d = 1.02$; and day 4, $p = 0.0.466$). **e** Bar graph displaying the latency to eat in vehicle- or CNO-treated AgRP-Gi mice ($n = 4$ and 5 mice per group; two-way ANOVA: $p$-time × treatment = 0.008, Cohen's $f = 0.861$; Sidak's post hoc test: vehicle- vs. CNO-treated AgRP-Gi mice on day 2, $p = 0.004$). **f** Bar graph displaying 1-hour food intake after saccharin exposure in vehicle- or CNO-treated AgRP-Gi mice ($n = 7$ and 6 mice per group; t-test: day 1, $p = 0.065$; day 2, $p = 0.027$, Cohen's $d = 1.46$; day 3, $p = 0.0.282$; and day 4, $p = 0.276$). **g** Representative photomicrographs of coronal section of ARH of AgRP-Gi mice treated with vehicle or CNO subjected to chromogenic immunohistochemistry against c-Fos (cyan) and fluorescent immunohistochemistry against AgRP (red). Colocalization is shown with white arrowheads. **h** Representative photomicrographs of coronal section of paraventricular hypothalamic nucleus (PVH) of AgRP-Gi mice treated with vehicle or CNO subjected to chromogenic immunohistochemistry against c-Fos (brown signal, blue arrowheads). Scale Bar: 50 μm low magnification and 25 μm high magnification. Data represent the mean ± SEM, with error bars representing the SEM. *, $p < 0.05$; †, $p < 0.1$ vs. different treatment; a, $p < 0.05$ vs day 0 same treatment.

that of WT mice, regardless of CNO treatment, as expected. Briefly, saccharin intake increased on day 2 of calorie restriction compared to day 0 and remained elevated on days 3 and 4 in both CNO- and vehicle-treated CR control mice, with no significant differences between groups. CNO- and vehicle-treated CR control mice were perfused on day 5 of the protocol, prior to saccharin exposure, and we found that the fraction of AgRP+ cells positive for c-Fos increased similarly in the ARH of both experimental groups (Supplementary Fig. 5).

In AgRP-Gi mice, BW decreased to a similar extent as in control mice (Supplementary Fig. 5) and was not affected by CNO treatment (Fig. 4c). To obtain a more comprehensive understanding of the effects of AgRP neuron inhibition on consummatory behaviors, we quantified not only saccharin intake but also latency to initiate eating and food intake 1 h after food presentation in both vehicle- and CNO-treated AgRP-Gi mice subjected to the calorie restriction and daily saccharin access protocol. We found that saccharin intake increased in vehicle-treated CR AgRP-Gi mice from day 2 to day 4 compared to day 0, whereas saccharin intake in CNO-treated CR AgRP-Gi mice increased only on days 3 and 4 (Fig. 4d). Notably, saccharin intake in CNO-treated CR AgRP-Gi mice on days 2 and 3 corresponded to ~28% and ~51%, respectively, of the intake observed in vehicle-treated CR AgRP-Gi mice (Fig. 4d). By day 4, no significant differences in saccharin intake were observed between groups. When analyzing food intake, we found that CNO-treated AgRP-Gi mice displayed a delayed latency to initiate eating on the experimental day 2 compared to vehicle-treated AgRP-Gi mice (Fig. 4e). Additionally, CNO-treated AgRP-Gi mice exhibited a reduction in food intake at 1 h after food exposure on experimental days 1 and 2 compared to vehicle-treated AgRP-Gi mice (Fig. 4f). To assess whether CNO treatment affects AgRP neuron activity in CR AgRP-Gi mice, we perfused vehicle- and CNO-treated mice on the fifth experimental day, prior to saccharin exposure, and performed double labeling for c-Fos and AgRP in their brain samples (Fig. 4g). We found that the fraction of AgRP+ cells positive for c-Fos in CNO-treated CR AgRP-Gi mice was reduced by ~42% compared to vehicle-treated CR AgRP-Gi mice (20 ± 7% vs. 47 ± 12%, t-test, $p = 0.043$, Cohen's $d = 1.23$). Analysis of AgRP+ signal in the ARH revealed that CNO-treated CR AgRP-Gi mice exhibited a significant reduction in the number of clearly identifiable AgRP+ cell bodies (26.9 ± 3.8 vs. 43.4 ± 6.4 cells/side/section, t-test, $p = 0.029$, Cohen's $d = 1.38$), and a trend toward a ~40% decrease in AgRP+ signal intensity (0.005 ± 0.001 vs. 0.009 ± 0.002 mean intensity/area, t-test, $p = 0.096$, Cohen's $d = 0.86$) compared to vehicle-treated mice. Neuroanatomical analysis of c-Fos immunostaining in the BNST, PVH, PVT, and LHA revealed similar numbers of c-Fos+ cells across all regions, except for the PVH (Fig. 4h), where CNO-treated CR AgRP-Gi mice showed a trend toward a ~65% decrease compared to vehicle-treated mice (31 ± 13 vs. 10 ± 1 cells/side/section, t-test, $p = 0.067$, Cohen's $d = 1.05$).

## Discussion

To our knowledge, this study identifies a novel role for AgRP neurons, showing that their activation potentiates consummatory responses to a rewarding, non-caloric stimulus during energy deficit states in mice.

AgRP neurons play a crucial role in the regulation of feeding behavior and energy homeostasis, with their activity being tightly controlled by neuroendocrine, metabolic, and autonomic feedback as well as food cues[31–33]. The activation of AgRP neurons impacts both feeding and non-feeding pathways. Indeed, activation of AgRP neurons promotes food intake when food is available but shifts energy metabolism and induces behavioral changes, including driving other motivated goal-directed behaviors and reducing anxiety-like behaviors in the absence of food[10,31,33,34]. Here, we provide the first evidence that AgRP neurons activation induces saccharin consumption in satiated mice and partially mediates the increase in saccharin consumption observed in CR mice. Thus, AgRP neurons not only promote general feeding behaviors aimed at consuming calorie sources but also selectively enhance the drive to consume highly rewarding stimuli, regardless of their caloric content. This suggests that AgRP neurons are instrumental in directing consumption toward rewarding food sources during energy deficits, optimizing feeding behavior based on palatability and potentially preparing the organism to prioritize calorie-rich resources when available.

Our study provides critical insights into the neurobiological mechanisms underlying the increase in saccharin intake observed in CR mice. We show that ARH-ablated mice, in which nearly all AgRP neurons as well as other ARH neurons are eliminated[12], exhibited a diminished calorie restriction-induced saccharin intake indicating that certain ARH neurons are essential for orchestrating this consummatory behavior during energy deficit. We also found that the majority of activated cells in the ARH of CR mice, as indicated by c-Fos staining, were AgRP neurons, and most AgRP neurons were activated under calorie restriction. Furthermore, saccharin intake in CR mice correlated with the fraction of activated AgRP neurons, which also showed increased *Agrp* gene expression and elevated AgRP peptide levels. Together, these findings indicate a strong association between AgRP neuron activation and saccharin intake in CR mice. Supporting a causal role for AgRP neurons in the consumption of rewarding stimuli, we found that pharmacogenetic activation of AgRP neurons in satiated mice is sufficient to increase saccharin intake. Conversely, pharmacogenetic inhibition of AgRP neurons in CR mice transiently reduced saccharin intake in the experimental days 2 and 3. These complementary approaches highlight the key role of the AgRP neuron activity enhancing the consumption of saccharin in CR mice. The more pronounced reduction in saccharin intake observed in CR ARH-ablated mice compared to CNO-treated CR AgRP-Gi mice may suggest that additional ARH neurons, beyond AgRP cells, contribute to this behavior. The reasons why CNO-treated calorie CR AgRP-Gi mice exhibit a transient reduction in saccharin intake remain uncertain.

Notably, CNO treatment—at the highest dose shown not to induce off-target effects[35]—resulted in a significant but partial reduction of calorie restriction-induced c-Fos increase in AgRP neurons on the fifth experimental day. This raises the possibility that residual AgRP neuron activity could still mediate saccharin intake during this period. Alternatively, the transient reduction in saccharin intake observed in CNO-treated CR AgRP-Gi mice may reflect the emergence of additional adaptations as the energy deficit becomes more pronounced over time. These mechanisms may involve a reorganization of neuronal systems—distinct from AgRP neurons—that regulate consummatory behavior to compensate for their pharmacological inhibition, as demonstrated in studies investigating AgRP neurons or other circuits[36,37]. The transient reduction in saccharin intake may also reflect a temporary influence of peripheral signals that enhance reward-related consummatory behavior via activation of AgRP neurons—such as ghrelin, which progressively increases with continued calorie restriction[38], requires AgRP neurons activation to induce food intake[5,39,40] (as we confirmed here), and potently affects consummatory behaviors[8]—but whose effects may later be overridden by additional signals arising from more severe energy deficits, such as hypoglycemia.

To identify brain targets that may contribute to enhanced saccharin intake in CR mice, we quantified c-Fos expression in regions known to mediate the consummatory responses of AgRP neurons[9,30]. We found increased numbers of activated cells in the PVH and BNST, with a trend toward increased activation in the LHA and PVT, suggesting that these brain centers are recruited by energy deficit. AgRP neurons are inhibitory GABA neurons; however, previous studies have shown that DREADD-induced activation of AgRP neurons elicits c-Fos expression in the PVH, LHA, BNST, and dorsal raphe nucleus[41]. Here, we observed that DREADD-induced inhibition of AgRP neurons tended to attenuate the increase in c-Fos expression in the PVH of CR mice, suggesting that this hypothalamic nucleus is part of the neuronal circuit through which AgRP neurons enhance saccharin intake during energy deficit. The molecular mechanisms by which calorie restriction-induced AgRP neurons activation recruits the PVH remain to be elucidated, but may involve not only AgRP itself, but also NPY and GABA, which modulate food intake over different timeframes[2,9,18,42,43]. For instance, AgRP neurons have been shown to activate corticotropin-releasing hormone neurons in the PVH by inhibiting presynaptic terminals from tonically active GABAergic afferents originating in the BNST[44]. Importantly, we cannot rule out the possibility that c-Fos induction in certain brain regions of CR mice reflects energy deficit–driven pathways independent of AgRP neurons, or that c-Fos lacks sufficient sensitivity to reliably indicate neuronal activation under our experimental conditions. For instance, projections from AgRP neurons to the LHA have been shown to enhance sweet taste palatability[45]; however, we observed only a trend toward increased c-Fos expression in the LHA of CR mice. Given that AgRP fiber density increases in specific brain areas in fasted mice[26], we also quantified it in the brain regions implicated in the orexigenic responses of these neurons[9,30]. We found that AgRP fiber density in the PVH and PVT of CR mice was unchanged, suggesting that changes in consummatory behavior during calorie restriction are not due to structural remodeling of AgRP projections in these regions. In contrast, AgRP fiber density in the BNST was decreased in CR mice compared to *ad libitum*-fed mice. The reason why calorie restriction selectively affects projections to this area remains uncertain, but it is intriguing that a recent study found that fasting-induced activation of AgRP neurons increases their synaptic connectivity to BNST neurons[46], suggesting that the reduced AgRP+ signal in the BNST may reflect increased neuropeptide release. Altogether, our findings support the conclusion that AgRP neuron activity—rather than remodeling of their projections—is the primary driver of enhanced consummatory behavior during calorie restriction although specific projections to the BNST may uniquely contribute to adaptive responses to energy deficit.

Here, we used a slightly modified version of a previously published protocol[47], in which saccharin intake is assessed prior to refeeding CR mice,

to estimate purely reward-related consummatory responses resulting from an energy deficit[1]. Early studies have shown that saccharin solutions, at certain concentrations, are rewarding to rodents and evoke phasic dopamine release in the nucleus accumbens core[48–50]. Since saccharin lacks calories, its gustatory-driving rewarding properties are independent of post-ingestive nutritional factors, which are known to contribute to the rewarding value of natural sugars through reinforcing gut-derived cues[27,51,52]. Thus, the use of saccharin as a consummatory stimulus allowed us to assess shifts in reward-related consummatory behavior without major interference from homeostatic components. Consistent with saccharin's rewarding properties, *ad libitum*-fed mice consumed it in high volumes and preferred it over water during overnight priming. To maximize our ability to detect subtle changes in reward-related consummatory responses, we assessed saccharin consumption prior to refeeding, during the light phase—when fluid consumption in mice is minimal and reward-related consummatory behavior appears to be preserved, as evidenced by operant conditioning studies assessing sucrose-seeking responses[53]. As expected, *ad libitum*-fed mice showed low saccharin intake, while CR mice exhibited increased intake and preference, suggesting that saccharin consumption reliably reflects energy deficit-enhanced reward behavior. Moreover, the marked difference in saccharin intake between *ad libitum*-fed and CR mice increases the likelihood of unmasking group differences, thereby improving the sensitivity of the protocol to detect experimental effects. Other methodological considerations warrant attention. For instance, the temporal window for testing saccharin intake—immediately preceding scheduled feeding—coincides with the period of food anticipatory activity, a well-established behavioral phenomenon characterized by heightened arousal and motivation[54]. This alignment may have contributed to increased exploratory behavior and enhanced contact with, and intake of, the saccharin solution, independent of its rewarding value, although it does not appear to have affected nonspecific consummatory behaviors, as water intake remained unchanged during testing. Moreover, repeated exposure across days could induce learning or expectation effects that influence intake beyond purely reward-related components[55]. Saccharin consumption is also subject to inter-individual variability due to differences in palatability preference[56], and may be influenced by nonspecific discomfort, such as that induced by IP injections used in some experiments. In addition, we observed some variability in daily saccharin intake across experiments, which may be attributed to a combination of factors, including intra-vivarium conditions, operator-related variability, and external influences such as seasonal changes[57–59]. Despite these considerations, the use of the calorie restriction and daily saccharin access protocol, along with the inclusion of appropriately control groups in each experimental design, proved valuable in uncovering important aspects of the neurobiological mechanisms underlying reward-related consummatory behaviors.

Furthermore, it is important to mention that the use of Cre/LoxP and DREADD technologies to manipulate AgRP neurons activity warrants careful consideration. In this study, we performed a neuroanatomical characterization of the AgRP-Cre mouse line and validated the functional activation or inhibition of AgRP neurons under our experimental conditions. However, some studies have reported hM4Di-mCitrine expression in Gi mice even in the absence of Cre activity, which could potentially influence our observations[20]. Additionally, it is important to note that DREADD-mediated modulation lacks temporal precision[60], and variations in experimental protocols for CNO administration affect the onset and duration of DREADD effects and consequently may lead to different outcomes. For instance, a recent study showed that AgRP-Cre mice with intra-ARH injections of hM3Dq-mCherry AAV virus did not increase saccharin consumption when CNO was administered immediately before testing[14]. Furthermore, variability in stereotactic AAV virus injections into the ARH may contribute to inconsistent results. Despite these considerations, the carefully performed behavioral and neuroanatomical approaches reported here in genetically modified mouse models strongly support a role for AgRP neurons in controlling consummatory behaviors driven by reward.

**Article**

In conclusion, the current study reveals a role for AgRP neurons in regulating consummatory behaviors toward a rewarding stimulus that lacks caloric value, a mechanism that becomes particularly relevant under conditions of energy deficit. This finding may have translational significance, as it could also contribute to the well-established phenomenon in which fasting biases reward systems toward high-caloric foods in humans[61,62]. Understanding the neurobiological basis by which AgRP neurons regulate consummatory behaviors offers promising avenues for developing treatments for eating disorders and obesity, enabling the targeted modulation of food reward without necessarily altering caloric intake.

## Reporting summary
Further information on research design is available in the Nature Portfolio Reporting Summary linked to this article.

## Data availability
All data supporting the findings of this study are available within the paper and its Supplementary Information and Supplementary Data. The numerical source data for graphs and charts are provided in the Supplementary Data file.

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

## Acknowledgements
This work was supported by grants from Fondo para la Investigación Científica y Tecnológica (FONCyT, PICT2019-3054 and PICT2020-3270 to MP, PICT2020-2692 to MPC, and PICT2020-1303 to GF), Fundação de Amparo à Pesquisa do Estado de São Paulo (FAPESP, 2016/17968-6 and 2021/14506-0 to RR) and the Novo Nordisk Foundation (Nr. 0092558 to HBS). We would like to thank to Cintia Bruno, Lucas Aguilar, Luján Gomez and Guadalupe García-Romero for their assistance on generating and controlling experimental mice.

## Author contributions
D.A.C., F.B., M.R., G.F., M.P.C., M.F.H, N.F. and H.J.F. carried out the experiments. D.A.C., F.B., G.F. and M.J.T performed the measurements, analysis and calculations. D.A.C., F.B., P.N.D.F. and H.B.S. contribute to data analysis and interpretations. R.R. and M.P designed, planned and supervised the work. D.A.C. and M.P wrote the main manuscript text. D.A.C prepared figures. All authors reviewed the manuscript.

## Funding

## Competing interests
The authors declare no competing interests.

## Ethics approval
All experiments received approval from the Institutional Animal Care and Use Committees of the IMBICE (21-0430 A) and the Federal University of São Paulo (12/2017).

## Consent to publish and consent to participate
The study does not involve human subjects and the manuscript contains any individual person's data in any form.
