## [Transparent Peer Review file · Communications Biology]

AgRP neuron activity enhances reward-related consummatory behaviors during energy deficit in mice

Corresponding Author: Dr MARIO PERELLO

Version 0:

Reviewer comments:

Reviewer #1

(Remarks to the Author)

Cassano et al demonstrate that increased consumption of saccharin during calorie restriction depends in part on AgRP neuron activity. They describe a series of well controlled calorie restriction experiments with various mouse models testing the involvement of the arcuate nucleus per se and AgRP neurons specifically in the escalation of saccharin intake as a consequence of energy deficit.

A limitation of these experiments is the timing of saccharin exposure during the light phase. Ad lib fed mice will be greatly inactive during this time and as a consequence less engaged with the saccharin bottle available. The authors use saccharin as a rewarding stimulus, but ad lib animals have a less than 50% preference for it (was the volume consumed recorded of the priming overnight exposure ?) and hardly any intake (presumably because they sleep most of the day). Calorie restricted mice develop food anticipatory activity before scheduled food ration and so have the opportunity to consume saccharin as seen in Figure 1 d and f escalating saccharin intake over the four days. That makes interpreting these results as a consequence of energy deficit per se difficult.

Moreover, the authors show immunohistochemical results from the activity marker cfos in AgRP target regions. Increased cfos in BNST, PVH, LHA and PVT can be a function of CR rather than caused by increased AgRP neuron activity, especially considering that AgRP neurons are exclusively inhibitory (GABA, NPY, AgRP all inhibit their target). In the literature is only a disinhibition of CRH neurons in the PVH by AgRP neurons described (<https://doi.org/10.1038/s41586-023-06358-0>) that could result in increased cfos in the PVH. Can the authors demonstrate that this is a more common mechanism how AgRP neurons control downstream neuron populations? I found the trending results for the LHA intriguing and it reminded me of results from Fu et al (<https://doi.org/10.1038/s41467-019-12478-x>) describing a hunger induced AgRP-> LHA VGlut2 pathway that modifies taste.

PVH cfos staining shown in Figure 4 h vehicle compares more to the fed control rather than the CR cfos stain shown in Fig 3B. Maybe animals were not that food restricted ? This could explain the non-significant difference in cfos.

Also, the square indicating the high magnification Fig 3B fed should be a bit lower over the heart shaped cfos positive neuron.

Alex Reichenbach

Reviewer #2

(Remarks to the Author)

This paper addresses the role of AgRP neurons in calorie restriction- induced increased intake in saccharin. It is well known that activation of AgRP neurons increases feeding behavior. The authors claim that they revealed a role of AGRP neurons in driving also feeding of non-caloric foods.

Major comments

It is uncertain whether these mice sense that saccharin has no calories. The authors should contrast saccharin in their model with sucrose. The mice may increase saccharin intake due to frustration of non-access to food.

I have ethic concerns that mice were treated with colchicine. A statement should be included on the extent of suffering and definition of human endpoints for the mice treated with colchicine. Perhaps as a warning that this may have been avoided. I have seen that without colchicine AgRP antibodies stain Arc cells.

The monosodium glutamate ablation should be introduced better. It is not clear enough why this experiment was necessary and what it precisely contributes

The results of fig 4 should be explained in more detail. I assume the same protocol was used again (?): The saccharin exposure from 10:00 to 14:00, followed by feeding with 40% of their average food intake, was repeated daily for 4 consecutive days. Why is saccharin intake much less than in figure 1? In fig 4f on day 4 inhibition of AgRP neurons does not seem to affect saccharin intake anymore. This suggests that AgRP neurons are not required for saccharin consumption? This goes against their earlier conclusion. Please clarify.

Minor comments

- What was the titer of the virus injected?
- When are lights on?
- Figure 3: AgRP neurons are GABAergic and CNO induced fos in projection areas. Add discussion that this may reflect disinhibition.
- Add discussion on why only in BNST AgRP staining decreases, whereas in other Fos-activated projection areas AgRP is not affected

Reviewer #3

(Remarks to the Author)

In this manuscript the authors show that AgRP cells in the ARH hold a critical role in driving intake of rewarding substances during energy deficit, independent of caloric content. They show that AgRP neurons in the ARH are both sufficient to induce increased intake of a non-caloric rewarding liquid (saccharin) with hM3Dq experiments, as well as partly necessary for the intake of saccharin with hM4Di experiments in mice. Thereby they show which AgRP-ARH target brain areas are possibly recruited during this process.

Overall the work looks thorough and convincing and the work is interesting and I enjoyed reading it, however it would be even more convincing if especially point 1 of my comments will be taken into consideration.

It is great that the authors included verification of the transgenic animals and viral injections, that gives a transparent and solid basis for the rest of the experiments.

Point 1. From the data it is not clear to me whether the water intake alone is altered parallel to saccharin during the saccharin tests. It would be nice to include data on only the water intake as well, to show whether the effect is selective for sweet saccharin solution or also observed in non-sweet water. It would also be interesting to know whether water intake increases in absence of saccharin solution.

Point 2. Maybe I have missed it, otherwise it would be good to include a reference to research that shows that saccharin is rewarding.

Point 3. A small detail that can be helpful, In the introduction it could be good to mention that saccharin is given as a solution.

Point 4. Are the WT mice C57Bl/6J or C57Bl/6N? And what background are the transgenic mice on? This is not clear to me from the manuscript.

Point 5. Regarding the feeding state of the mice before perfusion, I was wondering whether I understand correctly that part of the mice are fed, part is CR before perfusion? It could be helpful for readers to mention this explicitly in the 'calorie restriction protocol' (line 98-109).

Point 6. Do I understand correctly that the Gi mice receive the same dose of CNO as the Gq mice?

Point 7. When assessing overnight saccharin consumption, is water available as well? (line 120-122)

Point 8. If I am not wrong, PBS usually refers to phosphate buffered saline, not phosphate buffered solution.

Point 9. Line 387, I believe you can remove the word 'of'.

Point 10. Some comments regarding Figure 4; D: it might be interesting to consider whether the ghrelin levels on day 3 and 4. How would they compare to the ghrelin injection that you used? E: maybe you could use the word 'latency' on the y-axis? F: the title on the y-axis is not centered.

Point 11. Maybe I missed it, but I have not seen any images of the Gi DREADDs in the brain (mCitrine visualization).

Version 1:

Reviewer comments:

Reviewer #1

(Remarks to the Author)

The authors addressed completely and extensively my concerns .

Reviewer #2

(Remarks to the Author)

almost all my comments have been dealt with sufficiently, expect for the one on colchicine use. I propose the FOLLOWING ADDITION in the text:

To validate AgRP-cre line, AgRP-tdTomato mice (n=3) were intracerebroventricularly (ICV) injected with colchicine (8 µg/mouse, Sigma, C9754) in the lateral ventricle (coordinates: posterior=0.3 mm, lateral=1.0 mm, ventral=2.3 mm, from Bregma), as described previously²⁴. AS IT IS KNOWN THAT COLCHICINE TREATMENT CAN RESULT IN SEVERE DISCOMFORT, THIS TREATMENT WAS KEPT AS SHORT AS POSSIBLE AND EXTRA CARE WAS TAKEN TO AVOID SUFFERING. Mice were individually housed and monitored continuously to assess general behavior (activity, grooming, posture), food and water intake, and signs of distress (piloerection, vocalization, abnormal gait, hunched posture). No animals met the humane endpoint criteria—such as sustained loss of righting reflex, >20% BW loss, severe dehydration, lack of grooming, or persistent abnormal behavior—and all were perfused 48 h after colchicine injection. Brain sections were subjected to a fluorescent IHC anti-AgRP as described above.”

Reviewer #3

(Remarks to the Author)

No further comments, all my remarks have been satisfied.

Point by point response to reviewers' comments

Reviewer 1: Cassano et al demonstrate that increased consumption of saccharin during calorie restriction depends in part on AgRP neuron activity. They describe a series of well controlled calorie restriction experiments with various mouse models testing the involvement of the arcuate nucleus per se and AgRP neurons specifically in the escalation of saccharin intake as a consequence of energy deficit.

1. A limitation of these experiments is the timing of saccharin exposure during the light phase. Ad lib fed mice will be greatly inactive during this time and as a consequence less engaged with the saccharin bottle available. The authors use saccharin as a rewarding stimulus, but ad lib animals have a less than 50% preference for it (was the volume consumed recorded of the priming overnight exposure?) and hardly any intake (presumably because they sleep most of the day).

Response: We appreciate the reviewer's thoughtful observation regarding the timing of saccharin exposure. We fully agree with the reviewer that *ad libitum* fed mice are inactive during the light phase and as a consequence less prompt to perform any behavior, including consume food or saccharin. It must be highlighted, however, that the extent to which the light phase affects reward-related food intake is controversial topic and some studies found no effect of circadian rhythm on levels of active and inactive lever pressing, or lever discrimination for oral sucrose (doi:10.1016/j.bbr.2023.114650). In light of these findings and our more than 18 years of experience in the field of reward-related food intake, the experiments were purposefully conducted during the light phase. Specifically, our rationale was precisely to work in the light phase with low baseline saccharin intake, as can be appreciated in the group of *ad libitum* fed mice, to increase the sensitivity of our behavioral task to detect increments induced by energy deficit, and enhances our ability to reveal group differences (reductions in the CR-induced increase in saccharin intake) that might otherwise be masked by a ceiling effect in intake during the dark phase (when saccharin intake is much more elevated, please see our response to the comment #2 of the reviewer 3). We think that exposing animals to saccharin during the light phase provides a clearer readout of the motivational drive rather than simply capturing high activity or baseline intake patterns driven by circadian feeding rhythms.

Regarding the level of activity of mice when they were exposed to the saccharin bottle, it must be also considered that calorie restricted mice develop food anticipatory activity before scheduled foods, as highlighted by the reviewer in the following comment, and that this feature of the protocol needs to be taken in consideration.

Thanks for the comment regarding the use of saccharin as a rewarding stimulus and the observation that *ad libitum* animals drink small amounts of saccharin solution and show around 50% preference for it during the light phase — this is a valid point. Saccharin is rewarding for rodents, as saccharin is preferred over water and its consumption evokes phasic dopamine in the NAc core (10.1016/0006-8993(91)90946-s, 10.1002/syn.21519). Indeed, we found that mice in our experimental conditions consumed 0.76 ± 0.39 g of water and 4.41 ± 0.44 g of saccharin during the overnight priming exposure (dark phase), this information has been added to the manuscript. As the reviewer points out, however, both the absolute saccharin intake and the preference for it were substantially lower during the light phase. We believe this supports the rationale behind our experimental design, as it demonstrates that we are testing mice under conditions (i.e., the light phase) in which they naturally exhibit minimal consumption of both water and saccharin that remained stable across days. These low baseline levels of intake likely increased our sensitivity to detect differences between experimental groups.

To address the reviewer's concern, we have added the missing data to the Results section (i.e., water and saccharin intake during the overnight priming exposure, lines 230-231) and included a more detailed discussion in a dedicated paragraph, which is presented in our response to the next comment.

2. Calorie restricted mice develop food anticipatory activity before scheduled food ration and so have the opportunity to consume saccharin as seen in Figure 1 d and f escalating saccharin intake over the four days. That makes interpreting these results as a consequence of energy deficit per se difficult.

Response: We thank the reviewer for raising this important point. Yes, saccharin consumption was tested during the light phase, prior to food exposure, when mice typically exhibit food anticipatory activity (FAA) due to the scheduled feeding. We intentionally scheduled saccharin access to occur during this pre-meal window, which overlaps with the FAA period, as motivational processes related to reward-seeking are most likely to be heightened in calorie-restricted animals at this time. Importantly, we found that water intake did not change during the 4-hour saccharin access period, suggesting that the escalation in

saccharin intake more likely reflects increased motivation for rewarding stimuli under negative energy balance, rather than being solely a consequence of FAA. Nonetheless, we acknowledge that FAA may have contributed to the increased interaction with the saccharin bottle and have made readers aware of this possibility. To address this in the revised manuscript, we have added the missing data to the Results section (i.e., water intake during the protocol, lines 237-238 and 242-243) and discussed the potential contribution of FAA more explicitly as a co-occurring behavioral phenomenon. The paragraph addressing this and related concerns from other reviewers now reads: *“Here, we used a slightly modified version of a previously published protocol⁴⁷, in which saccharin intake is assessed prior to refeeding CR mice, to estimate purely reward-related consummatory responses resulting from an energy deficit¹. Early studies have shown that saccharin solutions, at certain concentrations, are rewarding to rodents and evoke phasic dopamine release in the nucleus accumbens core⁴⁸⁻⁵⁰. Since saccharin lacks calories, its gustatory-driving rewarding properties are independent of post-ingestive nutritional factors, which are known to contribute to the rewarding value of natural sugars through reinforcing gut-derived cues^{27,51,52}. Thus, the use of saccharin as a consummatory stimulus allowed us to assess shifts in reward-related consummatory behavior without major interference from homeostatic components. Consistent with saccharin's rewarding properties, ad libitum-fed mice consumed it in high volumes and preferred it over water during overnight priming. To maximize our ability to detect subtle changes in reward-related consummatory responses, we assessed saccharin consumption prior to refeeding, during the light phase—when fluid consumption in mice is minimal and reward-related consummatory behavior appears to be preserved, as evidenced by operant conditioning studies assessing sucrose-seeking responses⁵³. As expected, ad libitum-fed mice showed low saccharin intake, while CR mice exhibited increased intake and preference, suggesting that saccharin consumption reliably reflects energy deficit-enhanced reward behavior. Moreover, the marked difference in saccharin intake between ad libitum-fed and CR mice increases the likelihood of unmasking group differences, thereby improving the sensitivity of the protocol to detect experimental effects. Other methodological considerations warrant attention. For instance, the temporal window for testing saccharin intake—immediately preceding scheduled feeding—coincides with the period of food anticipatory activity, a well-established behavioral phenomenon characterized by heightened arousal and motivation⁵⁴. This alignment may have contributed to increased exploratory behavior and enhanced contact with, and intake of, the saccharin solution, independent of its rewarding value, although it does not appear to have affected*

nonspecific consummatory behaviors, as water intake remained unchanged during testing. Moreover, repeated exposure across days could induce learning or expectation effects that influence intake beyond purely reward-related components⁵⁵. Saccharin consumption is also subject to inter-individual variability due to differences in palatability preference⁵⁶, and may be influenced by nonspecific discomfort, such as that induced by IP injections used in some experiments. In addition, we observed some variability in daily saccharin intake across experiments, which may be attributed to a combination of factors, including intra-vivarium conditions, operator-related variability, and external influences such as seasonal changes⁵⁷⁻⁵⁹. Despite these considerations, the use of the calorie restriction and daily saccharin access protocol, along with the inclusion of appropriately control groups in each experimental design, proved valuable in uncovering novel aspects of the neurobiological mechanisms underlying reward-related consummatory behaviors.”

3. Moreover, the authors show immunohistochemical results from the activity marker cfos in AgRP target regions. Increased cfos in BNST, PVH, LHA and PVT can be a function of CR rather than caused by increased AgRP neuron activity, especially considering that AgRP neurons are exclusively inhibitory (GABA, NPY, AgRP all inhibit their target). In the literature is only a disinhibition of CRH neurons in the PVH by AgRP neurons described (<https://doi.org/10.1038/s41586-023-06358-0>) that could result in increased cfos in the PVH. Can the authors demonstrate that this is a more common mechanism how AgRP neurons control downstream neuron populations? I found the trending results for the LHA intriguing and it reminded me of results from Fu et al (<https://doi.org/10.1038/s41467-019-12478-x>) describing a hunger induced AgRP-> LHA VGlut2 pathway that modifies taste.

Response: We thank the reviewer for this insightful comment. We fully agree that AgRP neurons are predominantly inhibitory neurons. Strikingly, however, DREADD-induced activation of AgRP neurons has been shown to induce cFos in some of their targets, including not only the PVH, but also the lateral hypothalamic area, the bed nucleus of the stria terminalis and dorsal raphe nucleus (10.1016/j.cell.2016.02.044). Since AgRP neurons typically inhibit their postsynaptic targets, the activation of downstream regions such as the PVH may result from disinhibition, as elegantly shown for PVH neurons in the Douglass et al study cited by the reviewer.

In our study, we interpret the observed increases in cFos in BNST, PVH, LHA, and PVT as correlates of energy deficit-induced circuit engagement, rather than direct evidence

of AgRP-mediated excitation. However, we found that DREADD-induced inhibition of AgRP neurons tended to attenuate the CR-induced increase in c-Fos expression in the PVH suggesting that AgRP neurons activation is required for CR-induced c-Fos in the PVH.

We agree that the trending activation observed in the LHA is particularly intriguing. We appreciate the reviewer's reference to the work by Fu et al., which indeed highlights a functionally relevant AgRP→LHA VGlut2+ circuit that modulates taste sensitivity in the context of hunger. This study, among others, suggests that although AgRP neurons are inhibitory, their influence on downstream behavior can result from nuanced and circuit-specific disinhibition or modulation of excitatory populations. We now refer to this study in our revised Discussion to acknowledge this mechanism.

We have clarified in the revised manuscript that our cFos data provide evidence of brain regions recruitment in response to energy deficit, potentially influenced by AgRP neuronal activity in the PVH. We have also revised the Discussion to emphasize the importance of disinhibitory mechanisms in the interpretation of our results. The paragraph addressing this and related concerns from other reviewers now reads: *“To identify brain targets that may contribute to enhanced saccharin intake in CR mice, we quantified c-Fos expression in regions known to mediate the consummatory responses of AgRP neurons^{9,30}. We found increased numbers of activated cells in the PVH and BNST, with a trend toward increased activation in the LHA and PVT, suggesting that these brain centers are recruited by energy deficit. AgRP neurons are inhibitory GABA neurons; however, previous studies have shown that DREADD-induced activation of AgRP neurons elicits c-Fos expression in the PVH, LHA, BNST, and dorsal raphe nucleus⁴¹. Here, we observed that DREADD-induced inhibition of AgRP neurons tended to attenuate the increase in c-Fos expression in the PVH of CR mice, suggesting that this hypothalamic nucleus is part of the neuronal circuit through which AgRP neurons enhance saccharin intake during energy deficit. The molecular mechanisms by which calorie restriction-induced AgRP neurons activation recruits the PVH remain to be elucidated, but may involve not only AgRP itself, but also NPY and GABA, which modulate food intake over different timeframes^{2,9,18,42,43}. For instance, AgRP neurons have been shown to activate corticotropin-releasing hormone neurons in the PVH by inhibiting presynaptic terminals from tonically active GABAergic afferents originating in the BNST⁴⁴. Importantly, we cannot rule out the possibility that c-Fos induction in certain brain regions of CR mice reflects energy deficit-driven pathways independent of AgRP neurons, or that c-Fos lacks sufficient sensitivity to reliably indicate neuronal activation under our experimental conditions. For instance, projections from AgRP neurons to the LHA have been*

shown to enhance sweet taste palatability⁴⁵; however, we observed only a trend toward increased c-Fos expression in the LHA of CR mice. Given that AgRP fiber density increases in specific brain areas in fasted mice²⁶, we also quantified it in the brain regions implicated in the orexigenic responses of these neurons^{9,30}. We found that AgRP fiber density in the PVH and PVT of CR mice was unchanged, suggesting that changes in consummatory behavior during calorie restriction are not due to structural remodeling of AgRP projections in these regions. In contrast, AgRP fiber density in the BNST was decreased in CR mice compared to ad libitum-fed mice. The reason why calorie restriction selectively affects projections to this area remains uncertain, but it is intriguing that a recent study found that fasting-induced activation of AgRP neurons increases their synaptic connectivity to BNST neurons⁴⁶, suggesting that the reduced AgRP+ signal in the BNST may reflect increased neuropeptide release. Altogether, our findings support the conclusion that AgRP neuron activity—rather than remodeling of their projections—is the primary driver of enhanced consummatory behavior during calorie restriction although specific projections to the BNST may uniquely contribute to adaptive responses to energy deficit.”

4. PVH cfos staining shown in Figure 4 h vehicle compares more to the fed control rather than the CR cfos stain shown in Fig 3B. Maybe animals were not that food restricted? This could explain the non-significant difference in cfos.

Response: We thank the reviewer for this insightful observation and for their careful evaluation of the figures. The reviewer is absolutely correct—upon re-examination, we agree that the PVH c-Fos image in Figure 4h (vehicle-treated CR AgRP-Gi group) does not display the expected activation pattern typically observed under calorie restriction and is not representative of the overall findings for that experimental group. We apologize for this oversight and have now replaced the panel with a more representative image that more accurately reflects the c-Fos activation pattern observed in this group.

5. Also, the square indicating the high magnification Fig 3B fed should be a bit lower over the heart shaped cfos positive neuron.

Response: We thank the reviewer for this careful observation. The position of the square in Figure 3B (fed condition) has been adjusted accordingly to center it over the cFos-positive neuron of interest. The figure has been updated in the revised manuscript.

Reviewer 2: This paper addresses the role of AgRP neurons in calorie restriction- induced increased intake in saccharin. It is well known that activation of AgRP neurons increases feeding behavior. The authors claim that they revealed a role of AGRP neurons in driving also feeding of non-caloric foods.

1. Major comments

1.1. It is uncertain whether these mice sense that saccharin has no calories. The authors should contrast saccharin in their model with sucrose. The mice may increase saccharin intake due to frustration of non-access to food.

Response: We thank the reviewer for this insightful comment. We agree that it is important to consider the potential role of caloric feedback and learning processes in shaping reward-related consummatory behaviors. Although it may not be strictly necessary, we would like to emphasize that the rewarding properties of sugars arise from the integration of both gustatory and post-ingestive signals, whereas artificial sweeteners such as saccharin elicit reinforcement through gustatory input alone. The distinction between these pathways has been well characterized and is conserved across species. For instance, even in *Drosophila*, the neural circuits processing sweet taste and post-ingestive signals are distinct, as reviewed by de Araujo (Physiol Behav, 2016; doi:10.1016/j.physbeh.2011.04.039). Importantly, saccharin does not engage gut-derived reinforcing signals, which are a key contributor to the potent reinforcing effects of natural sugars. We intentionally used saccharin because it lacks caloric content, and its rewarding properties are independent of post-ingestive nutritional factors. Therefore, changes in its consumption primarily reflect shifts in reward-related consummatory behavior, without major interference from homeostatic components that would be introduced with sucrose. We have clarified these points in the revised Discussion section. The specific paragraph addressing this issue is included in our response to Comment 2 from Reviewer 1.

Regarding the last sentence of the comment, the increase in saccharin intake is likely to reflect primarily its gustatory-driven rewarding properties, but we cannot fully rule out that some component of increased saccharin intake may reflect food-seeking behavior or frustration due to restricted access to caloric sources. Nevertheless, we acknowledge the reviewer's point and have revised the manuscript to include the possibility that repeated exposure could lead to learning or expectation effects influencing intake beyond immediate reward, as follows. "*Moreover, repeated exposure across days could induce learning or expectation effects that influence intake beyond purely reward-related components* ⁵⁵".

1.2. I have ethic concerns that mice were treated with colchicine. A statement should be included on the extent of suffering and definition of human endpoints for the mice treated with colchicine. Perhaps as a warning that this may have been avoided. I have seen that without colchicine AgRP antibodies stain Arc cells.

Response: We thank the reviewer for raising this important ethical concern. We would like to emphasize that we are extremely careful and conscientious regarding animal welfare in all our experimental procedures. As the reviewer may have noticed, throughout the article we employed alternative mouse models to visualize AgRP neurons without the use of colchicine—namely, AgRP-TdTomato and NPY-GFP mice. However, as Reviewer #3 also acknowledged, we believed it was critical to independently validate the specificity and efficacy of the genetically-modified mouse lines used in our study. As shown in panel 2d, under our experimental conditions, AgRP neurons in *ad libitum*-fed mice are not clearly visualized, making reliable estimation of the overlap between AgRP+ cells and TdTomato+ cells difficult. To overcome this limitation, we used colchicine in a small cohort of AgRP-TdTomato mice to enhance AgRP detection in the soma and better estimate the degree of its colocalization with TdTomato signal. Importantly, the use of colchicine was approved by our institutional animal care and use committee. All efforts were made to minimize discomfort and distress: mice were closely monitored throughout the procedure, and humane endpoints were strictly defined and applied in full accordance with institutional ethical guidelines. We have now included the following detailed statement in the Methods section to clarify this point: “*To validate AgRP-cre line, AgRP-tdTomato mice (n=3) were intracerebroventricularly (ICV) injected with colchicine (8 µg/mouse, Sigma, C9754) in the lateral ventricle (coordinates: posterior=0.3 mm, lateral=1.0 mm, ventral=2.3 mm, from Bregma), as described previously²⁴. Mice were individually housed and monitored continuously to assess general behavior (activity, grooming, posture), food and water intake, and signs of distress (piloerection, vocalization, abnormal gait, hunched posture). No animals met the humane endpoint criteria—such as sustained loss of righting reflex, >20% BW loss, severe dehydration, lack of grooming, or persistent abnormal behavior—and all were perfused 48 h after colchicine injection. Brain sections were subjected to a fluorescent IHC anti-AgRP as described above.*”

1.3. The monosodium glutamate ablation should be introduced better. It is not clear enough why this experiment was necessary and what it precisely contributes.

Response: We thank the reviewer for this insightful comment. The use of ARH-ablated mice in our study was initially motivated by practical considerations. Specifically, these mice were employed at an early stage of the project to test the hypothesis that the ARH—the hypothalamic region containing AgRP neurons—contributes to the enhancement of consummatory responses to saccharin during energy deficit. At that time, ARH-ablated mice provided an accessible and well-characterized model for broadly disrupting ARH function. Upon obtaining supportive evidence using this approach, we implemented a more refined and cell-specific strategy by generating and utilizing AgRP-Cre mice and DREADDs.

Beyond their chronological role in the study design, ARH-ablated mice also offer complementary value to the experimental framework. These animals exhibit a broader and permanent disruption of the ARH, affecting not only AgRP neurons but also additional neuronal subtypes. This widespread lesioning results in a more pronounced and sustained reduction in CR-induced saccharin intake. In contrast, AgRP-Gi mice treated with CNO exhibit a selective—but transient—inhibition of AgRP neuron activity, leading to a temporary and less robust reduction in saccharin intake compared to vehicle-treated controls. The more evident impairment in CR-induced saccharin intake observed in ARH-ablated mice, as compared to CNO-treated AgRP-Gi mice, suggests that this behavior is governed by a broader hypothalamic network rather than AgRP neurons alone. Moreover, the transient nature of the behavioral suppression in the AgRP-Gi model, but not in the ARH-ablated mice, may indicate that compensatory mechanisms—either within the ARH neurons or neurons upstream or downstream of the ARH—may partially restore CR-driven saccharin intake even when AgRP neuron activity is inhibited. This compensatory capacity is likely absent or severely limited in ARH-ablated mice, where structural and functional integrity of multiple interacting circuits is permanently disrupted. Thus, the use of both models not only strengthens the overall conclusion that ARH integrity, in general, and AgRP neuron activation, in particular, are required to promote saccharin intake during calorie restriction, but also allows for a more nuanced understanding of the neural architecture supporting this behavior.

We have now revised the Results section to more clearly emphasize the pronounced and persistent impairment in CR-induced saccharin intake observed in ARH-ablated mice, compared to CNO-treated AgRP-Gi mice (Lines 250-252). Additionally, we have provided a more explicit explanation of the rationale and clarified the specific contribution of the ARH-ablated mice to the overall study in the revised version of the third paragraph of the discussion section, which now reads as follows: "*Our study provides critical insights into the*

neurobiological mechanisms underlying the increase in saccharin intake observed in CR mice. We show here that ARH-ablated mice, in which nearly all AgRP neurons as well as other ARH neurons are eliminated¹², exhibited a diminished calorie restriction-induced saccharin intake indicating that certain ARH neurons are essential for orchestrating this consummatory behavior during energy deficit. We also found that the majority of activated cells in the ARH of CR mice, as indicated by c-Fos staining, were AgRP neurons, and most AgRP neurons were activated under calorie restriction. Furthermore, saccharin intake in CR mice correlated with the fraction of activated AgRP neurons, which also showed increased *Agrp* gene expression and elevated AgRP peptide levels. Together, these findings indicate a strong association between AgRP neuron activation and saccharin intake in CR mice. Supporting a causal role for AgRP neurons in the consumption of rewarding stimuli, we found that pharmacogenetic activation of AgRP neurons in satiated mice is sufficient to increase saccharin intake. Conversely, pharmacogenetic inhibition of AgRP neurons in CR mice transiently reduced saccharin intake in the experimental days 2 and 3. These complementary approaches highlight the key role of the AgRP neuron activity enhancing the consumption of saccharin in CR mice. The more pronounced reduction in saccharin intake observed in CR ARH-ablated mice compared to CNO-treated CR AgRP-Gi mice may suggest that additional ARH neurons, beyond AgRP cells, contribute to this behavior. The reasons why CNO-treated calorie CR AgRP-Gi mice exhibit a transient reduction in saccharin intake remain uncertain. Notably, CNO treatment—at the highest dose shown not to induce off-target effects³⁵—resulted in a significant but partial reduction of calorie restriction-induced c-Fos increase in AgRP neurons on the fifth experimental day. This raises the possibility that residual AgRP neuron activity could still mediate saccharin intake during this period. Alternatively, the transient reduction in saccharin intake observed in CNO-treated CR AgRP-Gi mice may reflect the emergence of additional adaptations as the energy deficit becomes more pronounced over time. These mechanisms may involve a reorganization of neuronal systems—distinct from AgRP neurons—that regulate consummatory behavior to compensate for their pharmacological inhibition, as demonstrated in studies investigating AgRP neurons or other circuits^{36,37}. The transient reduction in saccharin intake may also reflect a temporary influence of peripheral signals that enhance reward-related consummatory behavior via activation of AgRP neurons—such as ghrelin, which progressively increases with continued calorie restriction³⁸, requires AgRP neurons activation to induce food intake^{5,39,40} (as we confirmed here), and potentially affects

consummatory behaviors⁸—but whose effects may later be overridden by additional signals arising from more severe energy deficits, such as hypoglycemia.

1.4. The results of fig 4 should be explained in more detail. I assume the same protocol was used again (?): The saccharin exposure from 10:00 to 14:00, followed by feeding with 40% of their average food intake, was repeated daily for 4 consecutive days. Why is saccharin intake much less than in figure 1? In fig 4f on day 4 inhibition of AgRP neurons does not seem to affect saccharine intake anymore. This suggests that AgRP neurons are not required for saccharin consumption? This goes against their earlier conclusion. Please clarify.

Response: We thank the reviewer for this insightful comment. The section entitled "Calorie restriction-induced enhancement of saccharin intake requires AgRP neuron activation", which includes the data presented in Figure 4, has now been expanded for clarity, and we hope it meets the reviewer's expectations. As correctly assumed by the reviewer, the same calorie restriction and daily saccharin access protocol was used throughout the manuscript—this has now been explicitly stated in the revised Results section to avoid any confusion. The text now reads as follows: "*To test whether AgRP neuron activation is required to enhance saccharin intake during energy deficit, we used DREADD technology to selectively inhibit AgRP neurons each day prior to saccharin exposure in CR mice. To generate mice in which AgRP neurons could be specifically inhibited, AgRP-Cre mice were crossed with Gi mice to produce AgRP-Gi mice, in which Cre-mediated removal of the floxed-STOP cassette induces selective expression of the inhibitory hM4Di-mCitrine in AgRP neurons. Here, we tested the specificity of hM4Di-mCitrine expression by analyzing mCitrine signal in the brain of CR mice and found it to be enriched in the ventromedial ARH, overlapping with AgRP signal in AgRP-Gi mice, whereas no such signal was observed in AgRP-Cre or Gi control mice (Supplementary Fig. 4). Also, we tested the effects of ghrelin in these mice pretreated with either vehicle or CNO to assess the functional efficacy of CNO-induced inhibition of AgRP neurons, given that systemic ghrelin treatment primarily increases food intake by acting on these neurons 7. We found that ghrelin increased food intake and c-Fos labeling in the ARH of vehicle- or CNO-treated WT, AgRP-Cre, and Gi mice, which were grouped together and referred to as "control" mice (Supplementary Fig. 4). Conversely, ghrelin increased food intake and c-Fos labeling in the ARH of vehicle-treated AgRP-Gi mice, but not in CNO-treated ones (Supplementary Fig. 4).*

Next, we subjected control and AgRP-Gi mice to the aforementioned calorie restriction and daily saccharin access protocol (Fig. 1a). In control mice, we found that their behavior in response to the protocol resembled that of WT mice, regardless of CNO treatment, as expected. Briefly, saccharin intake increased on day 2 of calorie restriction compared to day 0 and remained elevated on days 3 and 4 in both CNO- and vehicle-treated CR control mice, with no significant differences between groups. CNO- and vehicle-treated CR control mice were perfused on day 5 of the protocol, prior to saccharin exposure, and we found that the fraction of AgRP+ cells positive for c-Fos increased similarly in the ARH of both experimental groups (Supplementary Fig. 5).

In AgRP-Gi mice, BW decreased to a similar extent as in control mice (Supplementary Fig. 5) and was not affected by CNO treatment (Fig. 4c). To obtain a more comprehensive understanding of the effects of AgRP neuron inhibition on consummatory behaviors, we quantified not only saccharin intake but also latency to initiate eating and food intake 1 hour after food presentation in both vehicle- and CNO-treated AgRP-Gi mice subjected to the calorie restriction and daily saccharin access protocol. We found that saccharin intake increased in vehicle-treated CR AgRP-Gi mice from day 2 to day 4 compared to day 0, whereas saccharin intake in CNO-treated CR AgRP-Gi mice increased only on days 3 and 4 (Fig. 4d). Notably, saccharin intake in CNO-treated CR AgRP-Gi mice on days 2 and 3 corresponded to ~28% and ~51%, respectively, of the intake observed in vehicle-treated CR AgRP-Gi mice (Fig. 4d). By day 4, no significant differences in saccharin intake were observed between groups. When analyzing food intake, we found that CNO-treated AgRP-Gi mice displayed a delayed latency to initiate eating on the experimental day 2 compared to vehicle-treated AgRP-Gi mice (Fig. 4e). Additionally, CNO-treated AgRP-Gi mice exhibited a significant reduction in food intake at 1 hour after food exposure on experimental days 1 and 2 compared to vehicle-treated AgRP-Gi mice (Fig. 4f). To assess whether CNO treatment affects AgRP neuron activity in CR AgRP-Gi mice, we perfused vehicle- and CNO-treated mice on the fifth experimental day, prior to saccharin exposure, and performed double labeling for c-Fos and AgRP in their brain samples. We found that the fraction of AgRP+ cells positive for c-Fos in CNO-treated CR AgRP-Gi mice was reduced by ~42% compared to vehicle-treated CR AgRP-Gi mice ($20 \pm 7\%$ vs. $47 \pm 12\%$, t-test, $p < 0.05$; Fig. 4g). Analysis of AgRP+ signal in the ARH revealed that CNO-treated CR AgRP-Gi mice exhibited a significant reduction in the number of clearly identifiable AgRP+ cell bodies (26.9 ± 3.8 vs. 43.4 ± 6.4 cells/side/section, t-test, $p < 0.05$), and a trend toward a ~40% decrease in AgRP+

signal intensity (0.005 ± 0.001 vs. 0.009 ± 0.002 mean intensity/area, *t*-test, $p=0.0962$) compared to vehicle-treated mice. Neuroanatomical analysis of *c-Fos* immunostaining in the BNST, PVH, PVT, and LHA revealed similar numbers of *c-Fos*+ cells across all regions, except for the PVH, where CNO-treated CR AgRP-Gi mice showed a trend toward a ~65% decrease compared to vehicle-treated mice (31 ± 13 vs. 10 ± 1 cells/side/section, *t*-test, $p=0.067$; Fig. 4h)."

The reviewer is correct regarding the observed differences in saccharin intake between the vehicle-treated CR AgRP-Gi mice (Figure 4) and the CR WT mice (Figure 1), since the average intake is around 50% lower in the former group (5.6 ± 1.1 g vs. 2.2 ± 0.6 g on experimental day 4). We believe this difference may reflect the inherent variability in consummatory behaviors. Notably, only one out of five mice consumed more than 6 g/day of saccharin in the CR WT group, yet this single high value markedly influenced the group average. In contrast, none of the vehicle-treated CR AgRP-Gi mice reached that level. Variability in consummatory behavior is well documented in mice, and can be influenced not only by intra-vivarium factors but also by external conditions, such as seasonal changes, which can significantly affect both mice and humans. Additionally, specific aspects of the experimental design may subtly affect mouse behavior. For instance, in our case, it is possible that the intraperitoneal vehicle injections administered to control AgRP-Gi mice induced mild discomfort or stress, which may have modestly reduced saccharin intake compared to the WT mice, who did not receive injections. While such variability cannot be overlooked, it is important to emphasize that the relevant reference group for each comparison consists of the animals tested within the same experimental conditions — that is, those included in the statistical analysis. We now mention this caveat in the discussion section to alert readers to this potential confounding factor as follows: "Saccharin consumption is also subject to inter-individual variability due to differences in palatability preference⁵⁶, and may be influenced by nonspecific discomfort, such as that induced by IP injections used in some experiments. In addition, we observed some variability in daily saccharin intake across experiments, which may be attributed to a combination of factors, including intra-vivarium conditions, operator-related variability, and external influences such as seasonal changes⁵⁷⁻⁵⁹. Despite these considerations, the use of the calorie restriction and daily saccharin access protocol, along with the inclusion of appropriately control groups in each experimental design, proved valuable in uncovering novel aspects of the neurobiological mechanisms underlying reward-related consummatory behaviors."

Finally, the reviewer is right that that inhibition of AgRP neurons on day 4 did not significantly affect saccharin intake, as shown in Figure 4f. We have now addressed this point in the Discussion, where we offer potential explanations for this apparent dissociation, including possible habituation to the saccharin stimulus, compensatory mechanisms, or diminishing motivational drive as the protocol progresses. The added text reads: *" Our study provides critical insights into the neurobiological mechanisms underlying the increase in saccharin intake observed in CR mice. We show here that ARH-ablated mice, in which nearly all AgRP neurons as well as other ARH neurons are eliminated¹², exhibited a diminished calorie restriction-induced saccharin intake indicating that certain ARH neurons are essential for orchestrating this consummatory behavior during energy deficit. We also found that the majority of activated cells in the ARH of CR mice, as indicated by c-Fos staining, were AgRP neurons, and most AgRP neurons were activated under calorie restriction. Furthermore, saccharin intake in CR mice correlated with the fraction of activated AgRP neurons, which also showed increased *Agrp* gene expression and elevated AgRP peptide levels. Together, these findings indicate a strong association between AgRP neuron activation and saccharin intake in CR mice. Supporting a causal role for AgRP neurons in the consumption of rewarding stimuli, we found that pharmacogenetic activation of AgRP neurons in satiated mice is sufficient to increase saccharin intake. Conversely, pharmacogenetic inhibition of AgRP neurons in CR mice transiently reduced saccharin intake in the experimental days 2 and 3. These complementary approaches highlight the key role of the AgRP neuron activity enhancing the consumption of saccharin in CR mice. The more pronounced reduction in saccharin intake observed in CR ARH-ablated mice compared to CNO-treated CR AgRP-Gi mice may suggest that additional ARH neurons, beyond AgRP cells, contribute to this behavior. The reasons why CNO-treated calorie CR AgRP-Gi mice exhibit a transient reduction in saccharin intake remain uncertain. Notably, CNO treatment—at the highest dose shown not to induce off-target effects ³⁵—resulted in a significant but partial reduction of calorie restriction-induced c-Fos increase in AgRP neurons on the fifth experimental day. This raises the possibility that residual AgRP neuron activity could still mediate saccharin intake during this period. Alternatively, the transient reduction in saccharin intake observed in CNO-treated CR AgRP-Gi mice may reflect the emergence of additional adaptations as the energy deficit becomes more pronounced over time. These mechanisms may involve a reorganization of neuronal systems—distinct from AgRP neurons—that regulate consummatory behavior to compensate for their pharmacological inhibition, as demonstrated in studies investigating AgRP neurons or other*

circuits^{36,37}. The transient reduction in saccharin intake may also reflect a temporary influence of peripheral signals that enhance reward-related consummatory behavior via activation of AgRP neurons—such as ghrelin, which progressively increases with continued calorie restriction³⁸, requires AgRP neurons activation to induce food intake^{5,39,40} (as we confirmed here), and potently affects consummatory behaviors⁸—but whose effects may later be overridden by additional signals arising from more severe energy deficits, such as hypoglycemia.”

2. Minor comments

2.1. What was the titer of the virus injected?

Response: We thank the reviewer for this comment. The sentence now reads: "To pharmacogenetically activate AgRP neurons, anesthetized WT or AgRP-Cre mice were bilaterally injected in the ARH (posterior = 1.4 mm, lateral = ±0.3 mm, and ventral = 5.85 mm from Bregma) with 300 nL of pAAV8-hSyn-DIO-hM3D(Gq)-mCherry (Addgene; viral titer ~1.8×10¹³ vg/mL), as previously described."

2.2. When are lights on?

Response: Thank you for your question. We have now clarified this point in the manuscript. The text has been updated to read: "Mice were housed at 21±1°C with a 12-hour light/dark cycle (7:00 to 19:00) and had ad libitum access to chow and water, unless specified."

2.3. Figure 3: AgRP neurons are GABAergic and CNO induced fos in projection areas. Add discussion that this may reflect disinhibition.

Response: Thank you for this insightful comment. We agree with the reviewer and have addressed this possibility in the revised version of the Discussion section. Please see our response to comment 3 from Reviewer 1 for a detailed explanation and the updated paragraph discussing this topic.

2.4. Add discussion on why only in BNST AgRP staining decreases, whereas in other Fos-activated projection areas AgRP is not affected.

Response: The reason why calorie restriction selectively reduces AgRP immunoreactivity in the BNST, but not in other projection areas, remains uncertain. However, one plausible explanation is that this decrease reflects enhanced activity-

dependent neuropeptide release rather than reduced expression or innervation. Intriguingly, a recent study demonstrated that fasting-induced activation of AgRP neurons promotes increased synaptic connectivity specifically to BNST neurons (10.1038/s41467-024-49766-0, suggesting a dynamic and plastic response of this projection site to metabolic status. This enhanced connectivity may lead to greater AgRP release in the BNST during energy deficit, which could result in reduced detectable peptide levels due to increased turnover or depletion of the releasable pool. These findings support the idea that the BNST is a metabolically responsive target of AgRP neurons and may play a distinct role in the adaptive behavioral and physiological responses to negative energy balance. We now discuss this interpretation in the revised manuscript and have added the appropriate reference to support this point, as follows: *"To identify brain targets that may contribute to enhanced saccharin intake in CR mice, we quantified c-Fos expression in regions known to mediate the consummatory responses of AgRP neurons^{9,30}. We found increased numbers of activated cells in the PVH and BNST, with a trend toward increased activation in the LHA and PVT, suggesting that these brain centers are recruited by energy deficit. AgRP neurons are inhibitory GABA neurons; however, previous studies have shown that DREADD-induced activation of AgRP neurons elicits c-Fos expression in the PVH, LHA, BNST, and dorsal raphe nucleus⁴¹. Here, we observed that DREADD-induced inhibition of AgRP neurons tended to attenuate the increase in c-Fos expression in the PVH of CR mice, suggesting that this hypothalamic nucleus is part of the neuronal circuit through which AgRP neurons enhance saccharin intake during energy deficit. The molecular mechanisms by which calorie restriction-induced AgRP neurons activation recruits the PVH remain to be elucidated, but may involve not only AgRP itself, but also NPY and GABA, which modulate food intake over different timeframes^{2,9,18,42,43}. For instance, AgRP neurons have been shown to activate corticotropin-releasing hormone neurons in the PVH by inhibiting presynaptic terminals from tonically active GABAergic afferents originating in the BNST⁴⁴. Importantly, we cannot rule out the possibility that c-Fos induction in certain brain regions of CR mice reflects energy deficit-driven pathways independent of AgRP neurons, or that c-Fos lacks sufficient sensitivity to reliably indicate neuronal activation under our experimental conditions. For instance, projections from AgRP neurons to the LHA have been shown to enhance sweet taste palatability⁴⁵; however, we observed only a trend toward increased c-Fos expression in the LHA of CR mice. Given that AgRP fiber density increases in specific brain areas in fasted mice²⁶, we also quantified it in the brain regions implicated in the orexigenic responses of these neurons^{9,30}. We found that AgRP fiber density in the PVH and PVT of*

CR mice was unchanged, suggesting that changes in consummatory behavior during calorie restriction are not due to structural remodeling of AgRP projections in these regions. In contrast, AgRP fiber density in the BNST was decreased in CR mice compared to ad libitum-fed mice. The reason why calorie restriction selectively affects projections to this area remains uncertain, but it is intriguing that a recent study found that fasting-induced activation of AgRP neurons increases their synaptic connectivity to BNST neurons ⁴⁶, suggesting that the reduced AgRP+ signal in the BNST may reflect increased neuropeptide release. Altogether, our findings support the conclusion that AgRP neuron activity—rather than remodeling of their projections— is the primary driver of enhanced consummatory behavior during calorie restriction although specific projections to the BNST may uniquely contribute to adaptive responses to energy deficit.”

Reviewer 3: In this manuscript the authors show that AgRP cells in the ARH hold a critical role in driving intake of rewarding substances during energy deficit, independent of caloric content. They show that AgRP neurons in the ARH are both sufficient to induce increased intake of a non-caloric rewarding liquid (saccharin) with hM3Dq experiments, as well as partly necessary for the intake of saccharin with hM4Di experiments in mice. Thereby they show which AgRP-ARH target brain areas are possibly recruited during this process.

Overall, the work looks thorough and convincing and it the work is interesting and I enjoyed reading it, however it would be even more convincing if especially point 1 of my comments will be taken into consideration.

It is great that the authors included verification of the transgenic animals and viral injections, that gives a transparent and solid basis for the rest of the experiments.

1. From the data it is not clear to me whether the water intake alone is altered parallel to saccharin during the saccharin tests. It would be nice to include data on only the water intake as well, to show whether the effect is selective for sweet saccharin solution or also observed in non-sweet water. It would also be interesting to know whether water intake increases in absence of saccharin solution.

Response: Thank you for your important comment. You are absolutely right—this is relevant information that we omitted in the initial submission but clearly adds value to the interpretation of the results. We have now included the water intake data in the revised version to address whether the observed effects are specific to the saccharin solution or also extend to non-sweet water. As noted in our response to comment 2 from Reviewer 1 (see above), daily water intake did not change in either *ad libitum*-fed or calorie-restricted mice during the 4-hour period in which they had access to the saccharin bottle, prior to feeding. These data have now been included in the manuscript and discussed. Unfortunately, we did not conduct experiments in which CR mice were given access only to water during this 4-hour window, so we cannot assess whether water intake alone is affected under these conditions. Additionally, we could not find any published studies specifically evaluating changes in water intake during the food anticipatory activity period in CR mice subjected to scheduled feeding.

2. Maybe I have missed it, otherwise it would be good to include a reference to research that shows that saccharin is rewarding.

Response: We thank the reviewer for this important point. Indeed, multiple studies have shown that saccharin is rewarding to rodents. Even saccharin preference tests have been widely used as a behavioral measure of anhedonia in rodents (doi: 10.3389/fnbeh.2023.1143720), further supporting its relevance as a non-caloric reward stimulus. We have now added references demonstrating that saccharin can induce dopamine release in the nucleus accumbens, as well as reviews discussing the rewarding properties of artificial sweeteners and the brain networks involved in sugar-based flavor preferences. In response to this comment—as well as comments from Reviewer 1 (comments 1 and 2) and Reviewer 2 (comments 1 and 4)—we have revised the manuscript to better contextualize the behavioral relevance of saccharin in our model and to include the appropriate supporting literature. We appreciate the reviewer’s suggestion to strengthen this aspect of the discussion. The revised paragraph has been included in our response to Comment 2 of Reviewer 1.

3. A small detail that can be helpful, In the introduction it could be good to mention that saccharin is given as a solution.

Response: We thank the reviewer for this helpful suggestion. We have now clarified this detail in the Introduction. The revised sentence reads: "*To test this hypothesis, we employed an experimental paradigm in which mice were calorie-restricted (CR; i.e., fed 40% of their average daily food intake) and exposed daily to a non-caloric saccharin solution prior to feeding.*"

4. Are the WT mice C57BI/6J or C57BI/6N? And what background are the transgenic mice on? This is not clear to me from the manuscript.

Response: We thank the reviewer for this observation. All mice used in this study, including both wild-type and genetically modified animals, were on a C57BL/6J genetic background. We have clarified this in the Materials and Methods section by explicitly stating: “All genetically modified mice were backcrossed for more than 10 generations onto a C57BL/6J genetic background.”

5. Regarding the feeding state of the mice before perfusion, I was wondering whether I understand correctly that part of the mice are fed, part is CR before

perfusion? It could be helpful for readers to mention this explicitly in the ‘calorie restriction protocol’ (line 98-109).

Response: We thank the reviewer for this helpful suggestion and apologize for any misunderstanding. We have revised the description of the calorie restriction protocol to more clearly specify the feeding state of the mice prior to perfusion. The updated text now reads: "*Experimental design. The experimental design is shown in Figure 1a. Mice were individually housed in cages enriched with nesting material and shelters four days prior to the experiment, with ad libitum access to food and water to estimate their daily intake. The night before the experiment, they were given access to two drinking bottles: one containing a 0.1% sodium saccharin (Parafarm) solution and the other containing water. On the first experimental day, food was removed at 10:00, and mice were given 4-hour access to the saccharin solution while retaining access to water, the positions of the bottles (left or right side of the cage) were randomly assigned for each animal. Afterward, they were weighed and randomly assigned to experimental groups: either fed ad libitum (fed mice) or given 40% of their average daily intake (CR mice). This daily routine—saccharin exposure from 10:00 to 14:00 followed by either ad libitum or 40% feeding, depending on the group—was repeated for four consecutive days. Saccharin and water intake were calculated separately as raw values, and saccharin preference was calculated as (saccharin intake/total intake) × 100%, where total intake refers to the combined volume of saccharin and water consumed. On the fifth experimental day, both fed mice (which had been eating ad libitum) and CR mice (which had been receiving 40% of their average daily intake for the previous four days) were perfused at 10:00, prior to saccharin exposure.*"

6. Do I understand correctly that the Gi mice receive the same dose of CNO as the Gq mice?

Response: Yes, that is correct. Both AgRP-Gi and AgRP-Gq mice received the same dose of CNO. The text now specified the CNO dose in both cases.

7. When assessing overnight saccharin consumption, is water available as well? (line 120-122)

Response: Yes, water was available during the overnight saccharin consumption assessment. The text now reads: "*To assess saccharin consumption, AgRP-Cre mice injected with the hM3Dq-mCherry AAV virus were provided overnight access to two drinking bottles: one containing a 0.1% sodium saccharin solution and the other containing water*"

8. If I am not wrong, PBS usually refers to phosphate buffered saline, not phosphate buffered solution.

Response: We thank the reviewer for this correction. You are indeed correct—PBS is the standard abbreviation for phosphate-buffered saline. We have corrected this in the manuscript to reflect the accurate terminology.

9. Line 387, I believe you can remove the word ‘of’.

Response: Thank you for catching that. We have removed the word "of" from line 387 as recommended. Now, it reads: *“Notably, saccharin consumption exhibited dramatic increase in CR mice, compared to ad libitum-fed mice, highlighting its value to measure reward-related consumption, which is increased in energy deficit conditions.”*

10. Some comments regarding Figure 4; D: it might be interesting to consider whether the ghrelin levels on day 3 and 4. How would they compare to the ghrelin injection that you used? E: maybe you could use the word ‘latency’ on the y-axis? F: the title on the y-axis is not centered.

Response: We thank the reviewer for this insightful observation. We agree that endogenous ghrelin levels on days 3 and 4 may influence behavior and could be related to the decreased saccharin intake observed in CNO-treated AgRP-Gi mice, as shown in Fig. 4d. While we did not measure circulating ghrelin levels at these specific time points, this is an important consideration and could help further interpret the behavioral outcomes. We have now acknowledged this possibility in the revised discussion. We have now expanded the discussion as follow: *“The transient reduction in saccharin intake may also reflect a temporary influence of peripheral signals that enhance reward-related consummatory behavior via activation of AgRP neurons—such as ghrelin, which progressively increases with continued calorie restriction³⁸, requires AgRP neurons activation to induce food intake^{5,39,40} (as we confirmed here), and potently affects consummatory behaviors⁸—but whose effects may later be overridden by additional signals arising from more severe energy deficits, such as hypoglycemia.”*

The recommendations regarding panels E and F have been followed and the figure has been updated accordingly.

11. Maybe I missed it, but I have not seen any images of the Gi DREADDs in the brain (mCitrine visualization).

Response: We thank the reviewer for this observation. We have now included a supplementary figure (Supplementary Fig. 4) showing representative images of the ARH from AgRP-Gi, AgRP-Cre, and Gi-DREADD mice. As shown, mCitrine signal is enriched in the ventromedial ARH and overlaps with the endogenous AgRP signal specifically in AgRP-Gi mice, whereas no such signal is detected in AgRP-Cre or Gi-DREADD control mice